# WNT-induced association of Frizzled and LRP6 is not sufficient for the initiation of WNT/β-catenin signaling

Jan Hendrik Voss [1], Zsombor Koszegi [2,3], Yining Yan[1], Emily Shorter[4], Lukas Grätz [1], Johanna T. Lanner [4], Davide Calebiro [2,3] & Gunnar Schulte [1] ✉

The Wingless/Int-1 (WNT) signaling network is essential to orchestrate central physiological processes such as embryonic development and tissue homeostasis. In the currently held tenet, WNT/β-catenin signaling is initiated by WNT-induced recruitment of Frizzleds (FZDs) and LRP5/6 followed by the formation of a multiprotein signalosome complex. Here, we use bioluminescence resonance energy transfer (BRET) to show that different WNT paralogs dynamically trigger FZD-LRP6 association. While WNT-induced receptor interaction was independent of C-terminal LRP6 phosphorylation, it was allosterically modulated by binding of the phosphoprotein Dishevelled (DVL) to FZD. WNT-16B emerged as a ligand of particular interest, as it efficiently promoted FZD-LRP6 association but, unlike WNT-3A, did not lead to WNT/β-catenin signaling. Transcriptomic analysis further revealed distinct transcriptional fingerprints of WNT-3A and WNT-16B stimulation in HEK293 cells. Additionally, single-molecule tracking demonstrated that, despite increasing $FZD_5$ and LRP6 confinement, WNT-16B stimulation did not result in formation of higher-order receptor clusters, in contrast to WNT-3A. Our results suggest that FZD-WNT-LRP5/6 complex formation alone is not sufficient for the initiation of WNT/β-catenin signaling. Instead, we propose a two-step model, where initial ligand-induced FZD-LRP6 association must be followed by receptor clustering into higher-order complexes and subsequent phosphorylation of LRP6 for efficient activation of WNT/β-catenin signaling.

Wingless/Int-1 (WNT) lipoglycoproteins are morphogens of paramount importance in embryonic development, stem cell regulation, and tissue maintenance[1,2]. They initiate pleiotropic signaling cascades whose dysregulation results in pathologies such as diverse cancers and fibrosis, rendering components of the WNT signaling pathway attractive targets for therapeutic intervention[1,3]. WNT-dependent signaling cascades compose a tremendously complex network of pathways encompassing nineteen WNTs, their primary receptors (ten FZD paralogues, members of the class F of the G protein-coupled receptor superfamily), and diverse coreceptors (e.g., low-density lipoprotein receptor-related protein 5 and 6 (LRP5/6), receptor tyrosine kinase-like orphan receptor 1/2 (ROR1/2))[4]. On the intracellular side, the signal is

---

[1]Karolinska Institutet, Department of Physiology & Pharmacology, Sec. Receptor Biology & Signaling, Biomedicum S-17165 Stockholm, Sweden. [2]Department of Metabolism and Systems Science, College of Medicine and Health, University of Birmingham, Birmingham B15 2TT, UK. [3]Centre of Membrane Proteins and Receptors (COMPARE), Universities of Nottingham and Birmingham, Birmingham B15 2TT, UK. [4]Karolinska Institutet, Department of Physiology & Pharmacology, Sec. Molecular Muscle Physiology & Pathophysiology, Biomedicum S-17165 Stockholm, Sweden. ✉e-mail: gunnar.schulte@ki.se

propagated by two main downstream transducers, namely the phosphoprotein Disheveled (DVL) and heterotrimeric G proteins[4–6]. These components specify signaling that is characterized by the involvement of the transcriptional regulator β-catenin as either β-catenin-dependent (WNT/β-catenin signaling) or β-catenin-independent (e.g., planar-cell-polarity-like signaling). Signaling is initiated by WNT binding to the extracellular cysteine-rich domain (CRD) of FZDs. Simultaneous WNT binding to co-receptors contributes to signal specification and diversification, yet the molecular details remain poorly understood for most pathways. In WNT/β-catenin signaling, a signalosome model has emerged as a prevailing theory in the field[1,3,7].

According to this model, WNT binding to FZD and the co-receptors LRP5/6 leads to the clustering of multiple FZD-WNT-LRP5/6 complexes, scaffolded by DVL, at the plasma membrane, forming the so-called signalosome[7]. DVL condensates then serve as a docking platform for the multiprotein β-catenin destruction complex that constitutively ubiquitinates the transcriptional regulator β-catenin, dedicating it for proteasomal degradation[3,8]. In presence of WNTs, the destruction complex is disassembled and inactivated, and several components are recruited to the signalosome. As a result, β-catenin is stabilized and translocates to the nucleus, where it acts as a transcriptional regulator in concert with T-cell factor/lymphoid enhancer factor (TCF/LEF) transcription factors to induce the transcription of β-catenin target genes. This is accompanied by multiple biochemical changes, most prominently C-terminal phosphorylation of LRP6 and hyperphosphorylation of DVL.

More recently, synthetic activators of WNT/β-catenin signaling, known as WNT surrogates, were developed[9–11]. These surrogates mimic WNTs by binding simultaneously to FZD CRDs and LRP5/6, reinforcing the prevailing hypothesis that ligand-induced association of FZDs and LRP5/6 is not only necessary but also sufficient to initiate WNT/β-catenin signaling. Yet, due to the phylogenetic diversity of the 19 WNTs and ten FZD paralogs, it is reasonable to speculate that there are intricacies to the initiation of WNT/β-catenin signaling beyond purely crosslinking FZD and LRP5/6. In fact, FZD-WNT fusion proteins displayed FZD paralog-dependent differences in a WNT/β-catenin signaling output in vivo[12]. Further work unveiling essential conformational dynamics of the FZD core hinted at a more complex interplay of membrane proteins[13–18].

In this work, we establish a BRET assay between FZD and LRP5/6 to monitor the dynamic WNT-induced association between FZDs and LRP5/6. We demonstrate that FZD-LRP6 association is independent of LRP6 phosphorylation but is allosterically modulated by DVL. When probing WNT-FZD-LRP6 selectivity, we discovered that WNT-16B induces association of FZDs and LRP6 without triggering hallmarks of WNT/β-catenin signaling, such as LRP6 phosphorylation, higher-order receptor cluster formation, and TCF/LEF reporter gene activity. Based on these findings, we propose a model in which ligand-induced association of LRP6 with FZDs represents only the first step in signal initiation and is not necessarily followed by WNT/β-catenin signaling, which additionally requires clustering of FZD and LRP6 into higher-order clusters and phosphorylation of LRP6.

## Results

### A direct NanoBRET assay to capture dynamic association of FZD$_5$ and LRP6

To characterize the interactions between FZDs and LRP5/6 upon stimulation with WNTs, we established a direct NanoBRET assay using a C-terminally located Nanoluciferase (Nluc)/Venus system (Fig. 1A+B). For validation of the assay, we utilized FZD$_5$, which robustly mediates WNT/β-catenin signaling[14,19,20]. Surface expression of all constructs used in BRET assays was analyzed by surface ELISA (Supp. Figure 1A +B). We could detect all FZD-Nluc/Venus and LRP-Nluc/Venus constructs at the cell surface with limited surface expression of LRP5 constructs and FZD$_5$-Venus, whose expression levels were below the

detection threshold of the ELISA. LRP6 constructs required the co-expression of the chaperone MESD[21] for significant surface expression levels. Therefore, MESD was co-transfected in all experiments that included LRP5/6. Notably, co-expression of LRP6-Venus and FZD$_5$-Nluc decreased the surface expression of the latter, which nevertheless remained above the detection threshold (Supp. Figure 1C). We additionally confirmed functional responses to WNT-3A of all engineered FZD and LRP construct used throughout the study in TOPFlash reporter gene assays (Supp. Figure 1D+E).

In acceptor titration NanoBRET experiments, we did not observe constitutive interactions between FZD$_5$-Nluc and LRP6-Venus (Supp. Figure 2A). Given that WNT-3A is known to elicit stabilization of β-catenin by recruiting both FZD$_5$ and LRP6, we next tested its effect in ligand stimulation experiments. Here, we observed a robust and dynamic BRET increase between FZD$_5$-Nluc and LRP6-Venus upon addition of recombinant, purified WNT-3A in HEK293 cells, and a notably lower BRET increase between FZD$_5$-Nluc and LRP5-Venus, potentially due to lower surface expression of LRP5-Venus (Fig. 1C, Supp. Figure 1B). The assay displayed similar results with an inverse probe setup using LRP5/6-Nluc and FZD$_5$-Venus (Supp. Figure 2B). We decided to continue with FZD-Nluc/LRP5/6-Venus probes since FZD$_5$-Venus showed low surface expression levels (Supp. Figure 1A). A dynamic BRET response was not observed upon stimulation of an unrelated GPCR (β$_2$-adrenoceptor) and a CRD-truncated FZD$_5$ construct (ΔCRD-FZD$_5$) in combination with LRP6, confirming assay specificity (Supp. Figure 2C). A potential impact of endogenously expressed LRP5/6 was addressed by performing the experiments in HEK293T ΔLRP5/6 cells (Fig. 1D). Kinetic traces obtained here closely resembled those obtained in regular HEK293 cells emphasizing that endogenously expressed LRP5/6 do not contribute to the BRET read-out. Along these lines, stimulation with a WNT surrogate[9] resulted in a strong increase in ΔBRET between FZD$_5$-Nluc and LRP6-Venus, and to a lesser extent between FZD$_5$ and LRP5-Venus, comparable to what was observed in response to WNT-3A stimulation (Fig. 1E).

LRP5 and LRP6 transduce WNT signals with different efficacies, which was pinpointed to be a property of their C-terminal portions[22]. As LRP5 displayed lower ΔBRET values, we reasoned that exchanging the LRP5/6 C-termini might change the magnitude of the response to that of the respective other paralog. However, the traces observed in BRET experiments using FZD$_5$-Nluc and an LRP6 chimera with an LRP5 C-tail (LRP6-5CT-Venus) or vice versa (LRP5-6CT-Venus) were practically unchanged compared to traces obtained with the respective wild-type LRP5/6-Venus (Fig. 1F). Notably, the cell surface expression of LRP5-Venus and LRP5-6CT-Venus did not surpass the detection threshold, while LRP6-5CT-Venus retained the approximate surface expression of LRP6-Venus (Supp. Figure 1B). Our results suggest that the observed difference in ΔBRET between the traces of LRP5-Venus and LRP6-Venus did not result from a different conformational space obtained by their respective C-termini, but either from different expression levels or from their respective N-terminal/transmembrane domains.

Due to the lower surface expression of LRP5, we continued our experiments exclusively with LRP6. As WNT stimulation caused a dynamic BRET increase and constitutive interactions were not detectable, we conclude that our BRET assay measures WNT-induced association of FZD$_5$ and LRP6 rather than conformational dynamics of a pre-formed complex. Our findings contrast a study employing similar BRET probes that claimed a constitutive, but not dynamic interaction between mouse FZD$_8$ and LRP6[23].

### LRP6 phosphorylation is not required for WNT-induced FZD$_5$-LRP6 association

WNT-induced association of LRP5/6 and FZD is a hallmark of WNT/β-catenin signaling. In the process of WNT/β-catenin signaling, five C-terminal PPP(S/T)P-motifs of LRP6 are phosphorylated by GSK3 and

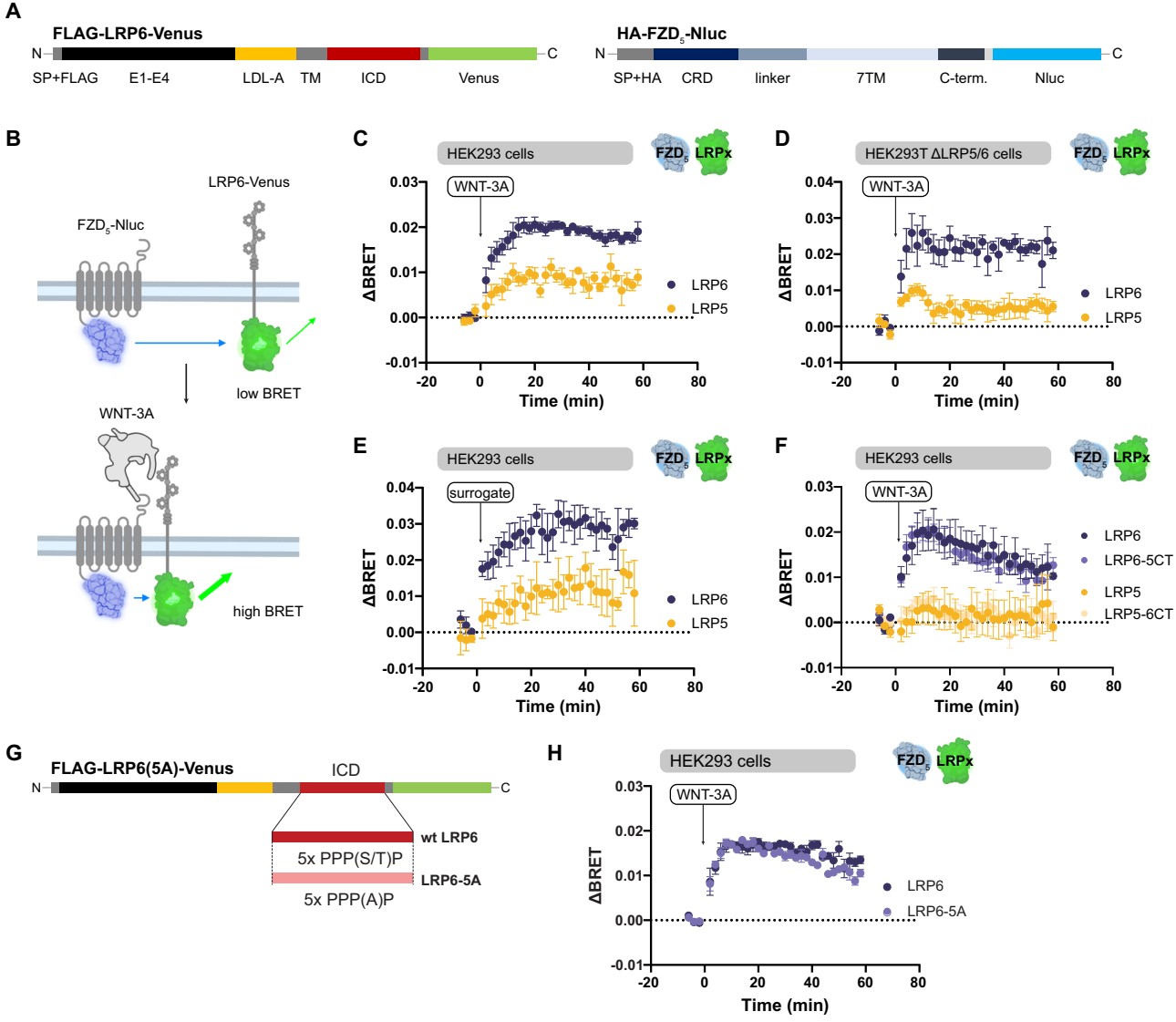

**Fig. 1 | Establishing a direct BRET assay to capture the dynamics in the WNT-3A-induced complex formation between FZD₅-Nluc and LRP5/6-Venus.**
**A** Architecture of LRP-Venus and FZD-Nluc constructs. SP, signal peptide; FLAG, FLAG-tag; E1-E4, extracellular YWTD β-propeller/EGF domain repeats 1-4; LDL-A, low-density lipoprotein receptor type A repeats; TM, transmembrane domain; ICD, intracellular domain; CRD, cysteine-rich domain; 7TM, seven-transmembrane domain; C-term., receptor C-terminal domain. **B** Schematic depiction of the assay principle. Parts of this figure were created with BioRender.com released under a Creative Commons Attribution-NonCommercial-NoDrivs 4.0 International license. Created in BioRender. Voss, J. https://BioRender.com/lfyvjb8. **C**, **D**. WNT-3A stimulation (1000 ng/ml)
kinetics of HEK293 (**C**) or HEK293T ΔLRP5/6 (**D**) cells transfected with FZD₅-Nluc and LRP5/6-Venus (yellow and blue circles, respectively; note: y-axes are not scaling equal in all figure panels). **E** Stimulation of HEK293 cells transfected with FZD₅-Nluc and LRP5- or LRP6-Venus with 1 nM of WNT surrogate. **F** WNT-3A stimulation (1000 ng/ml) of HEK293 cells transfected with FZD₅-Nluc and chimeric LRP5/6-Venus variants. **G** Architecture of LRP6 and LRP6-5A-Venus constructs. PPP(S/T)P/PPPAP, phosphorylation motifs. H. WNT-3A stimulation kinetics of HEK293A cells transfected with FZD₅-Nluc and LRP6-/LRP6-5A-Venus (blue and light blue circles, respectively). Data are presented as mean ± standard error of the mean (SEM) of five (**C**), four (**D**), or three (**E**, **F**, **H**) individual experiments, each performed in triplicate.

CK1 isoforms, components of the β-catenin destruction complex, presenting another proximal hallmark of the activation of the WNT/β-catenin pathway[24]. To determine whether LRP6 phosphorylation has an impact on receptor association as measured in the direct BRET assay, we generated an LRP6-Venus mutant lacking all phosphorylation sites in the C-tail (LRP6-5A-Venus; Fig. 1G)[25]. The mutant LRP6-5A-Venus expressed at the cell surface to a degree comparable to wild-type (WT) LRP6-Venus (Supp. Figure 1B). WNT-3A-stimulated BRET traces of FZD₅ and LRP6-5A-Venus behaved identical to WT LRP6-Venus (Fig. 1H), emphasizing that FZD₅-LRP6 association is independent of LRP6 phosphorylation.

To confirm signaling deficiency of the LRP6-5A mutant, we cloned a C-terminally untagged version of WT LRP6 and LRP6-5A. In agreement with the literature, LRP6-5A was in fact β-catenin signaling-

incompetent both by overexpression and in response to WNTs, as assessed by TOPFlash reporter gene assays (Supp. Figure 3)[25,26]. In contrast to native LRP6, recombinant expression of LRP6-5A could not rescue TOPFlash reporter gene activity upon WNT-3A stimulation in ΔLRP5/6 cells and acted in a dominant-negative fashion when expressed in HEK293 cells, probably due to outcompeting endogenous LRP6. In summary, our data suggest that the phosphorylation status of the LRP6 C-terminus affected neither the WNT-3A-induced FZD₅-LRP6 association nor its C-terminal conformation, despite significantly different signaling outputs.

## DVL modulates the conformation of the FZD₅-LRP6 complex
The phosphoprotein DVL is the main intracellular transducer of WNT-induced and FZD-mediated signaling and binds to FZDs with high

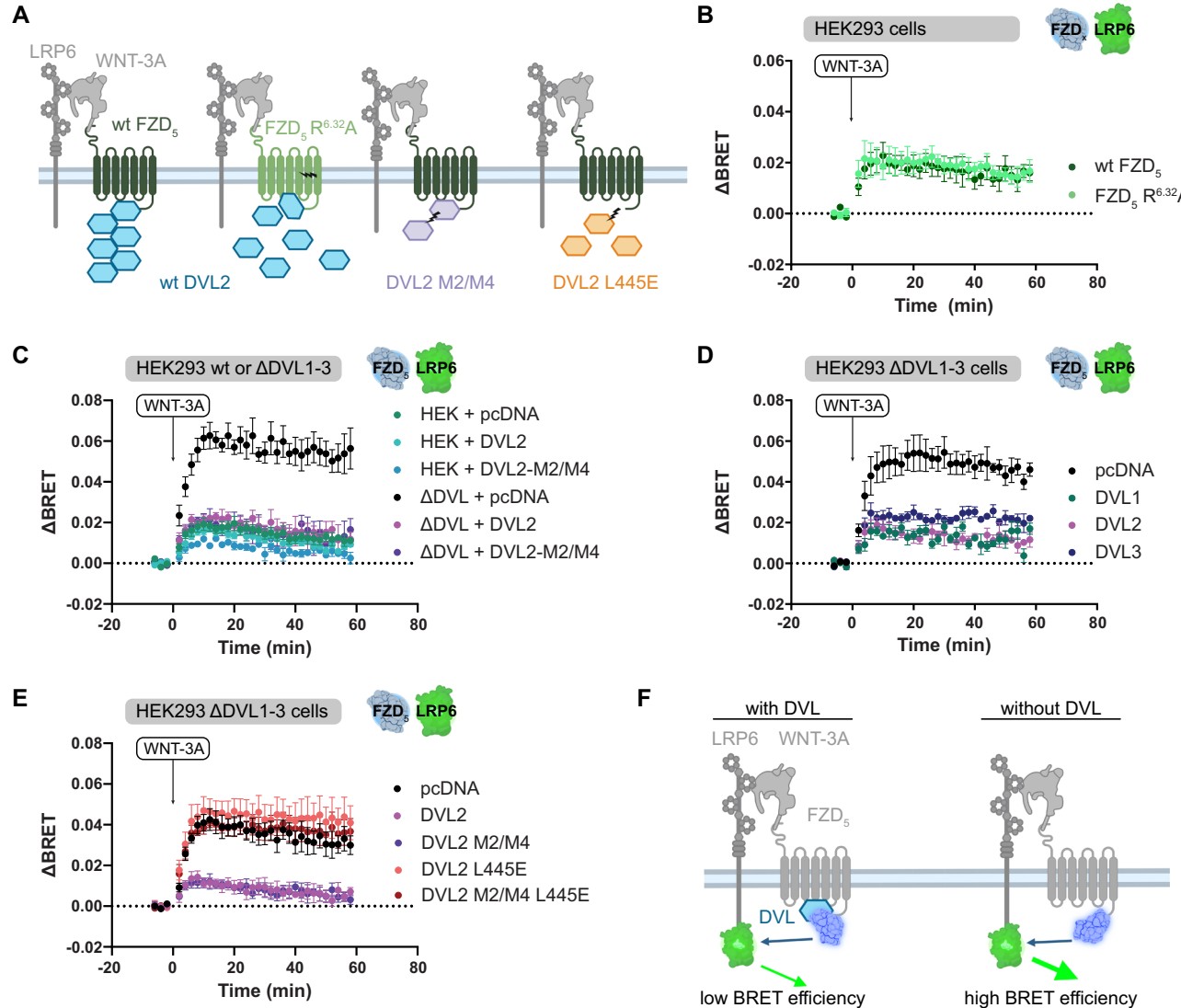

**Fig. 2 | DVL modulates WNT-induced FZD$_5$-LRP6 association. A** Schematic depiction of mutational paradigms. The FZD$_5$ R$^{6.32}$A mutants preferentially signals in DVL-independent fashion. The DVL2 M2/M4 mutant cannot polymerize by its DIX domains. The DVL2 L445E mutant cannot engage FZD via its DEP domain. Created in BioRender. Voss, J. (2025) https://BioRender.com/zfu7r84. **B** ΔBRET traces of HEK293 cells transfected either with WT FZD$_5$-Nluc or FZD$_5$ R$^{6.32}$A-Nluc (dark and light blue circles, respectively) and LRP6-Venus, stimulated with 1000 ng/ml WNT-3A. **C** ΔBRET traces of HEK293 and HEK293T ΔDVL1-3 cells, transfected with FZD$_5$-Nluc, LRP6-Venus and either pcDNA (green and black circles), DVL2 (cyan and magenta), or DVL2-M2/M4 (dark cyan and purple) as indicated, stimulated with 1000 ng/ml WNT-3A. **D** ΔBRET traces of ΔDVL1-3 cells, transfected with FZD$_5$-Nluc, LRP6-Venus and either pcDNA (black), DVL1 (green), DVL2 (magenta), or DVL3 (purple) as indicated, stimulated with 1000 ng/ml WNT-3A. **E** ΔBRET traces of ΔDVL1-3 cells, transfected with FZD$_5$-Nluc, LRP6-Venus and either pcDNA (black), DVL2 (magenta), DVL2-M2/M4 (purple), DVL2-L445E (orange), or DVL2-M2/M4-L445E (red) as indicated, stimulated with 1000 ng/ml WNT-3A. **F** Scheme illustrating a proposed mechanism of DVL modulation of WNT-induced FZD$_5$-LRP6 interaction. Created in BioRender. Voss, J. (2025) https://BioRender.com/3gct1ez. Data are shown as mean ± SEM of three (B – FZD$_5$ R$^{6.32}$A, C – data with DVL2-M2/M4 co-expression D, E) or four (B – wt FZD$_5$, C – all other data) individual experiments performed in triplicate. Parts of this figure were created with BioRender.com released under a Creative Commons Attribution-NonCommercial-NoDrivs 4.0 International license.

affinity mainly via its Disheveled, Egl-10 and Pleckstrin (DEP) domain[17,20]. DVL oligomers, polymerized by their Disheveled/Axin (DIX) domains, are an integral part of the WNT receptor signalosome where they mediate interactions between the upstream WNT receptors and proteins of the β-catenin destruction complex. To assess the potential impact of DVL on the interaction between FZD$_5$-Nluc and LRP6-Venus, we have analyzed WNT-3A-induced BRET responses in mutational paradigms from two angles (Fig. 2A): (i) a receptor-based angle, where we have employed the FZD$_5$ R$^{6.32}$A mutant, which prefers G protein- over DVL-mediated signaling[16], (ii) a DVL-based angle employing HEK293 ΔDVL1-3 cells in combination with non-functional mutants of DVL2, i.e. the oligomerization-deficient M2/M4 mutant[27,28] as well as the L445E mutant incapable to bind to FZDs[17,29].

In an initial experiment, we compared ΔBRET traces between LRP6-Venus and WT FZD$_5$-Nluc or FZD$_5$-Nluc R$^{6.32}$A upon WNT-3A stimulation and found no substantial difference between the traces (Fig. 2B). This suggested to us that DVL binding by FZD$_5$ would not modulate the WNT-3A-induced complex formation between FZD$_5$-Nluc and LRP6-Venus. However, when comparing the ΔBRET traces of WNT-3A-stimulated WT FZD$_5$-Nluc and LRP6-Venus between regular HEK293 cells and ΔDVL1-3 cells, the ΔBRET$_{max}$ increased substantially when DVL was absent (Fig. 2C). As DVL supposedly serves as a platform allowing clustering of multiple FZD$_5$-WNT-LRP6 complexes into higher-order complexes, the observed ΔBRET increase in ΔDVL1-3 cells came unexpected. We initially confirmed that both knockout and heterologous expression of DVL had no impact on FZD$_5$ and LRP6

surface expression (Supp. Figure 4A, B). The increased ΔBRET in ΔDVL1-3 cells could, however, be lowered to a level similar to that observed in native HEK293 cells by co-transfection of DVL2. This suggested that the observed phenomenon is based on presence of DVL2 instead of a cell-line specific difference (Fig. 2C, Supp. Figure 4C). Transfection of the oligomerization-deficient DVL2 M2/M4 mutant into HEK293 ΔDVL1-3 cells showed the same effect as transfection of WT DVL2. DVL2 therefore modulates the interaction between $FZD_5$-Nluc and LRP6-Venus in a DIX polymerization-independent manner. Similarly, co-transfection of DVL1 or DVL3 led to similar reductions in $\Delta BRET_{max}$ (Fig. 2D, Supp. Figure 4D) indicating that there are no DVL paralog-specific effects.

Similar observations were made in a combined approach using HEK293 ΔDVL1-3 cells and the $FZD_5$ $R^{6.32}A$ mutant, whereby ΔBRET between $FZD_5$ $R^{6.32}A$-Nluc and LRP6-Venus was increased in ΔDVL1-3 cells and could be lowered by recombinant DVL expression (Supp. Figure 4E+F). The $\Delta BRET_{max}$ observed for WT $FZD_5$ and the $R^{6.32}A$ mutant in ΔDVL cells was not significantly different from each other, suggesting that WNT-induced $FZD_5$-LRP6 complex formation is similarly affected by the absence of DVL1-3 for both WT $FZD_5$ and $FZD_5$ $R^{6.32}A$ (Supp. Figure 4G). Lastly, we utilized the DVL2 L445E mutant, whose DEP domain cannot interact with FZDs[17,29], and a combined DVL2 M2/M4 L445E mutant. From this, we found that neither of the DVL2 L445E mutants could dampen the increased ΔBRET in ΔDVL1-3 cells, emphasizing that the FZD-DVL interaction is the main driver of the difference in WNT-3A-induced $\Delta BRET_{max}$ between HEK293 and ΔDVL cells (Fig. 2E, Supp. Figure 4H).

In conclusion, the presence of FZD-binding DVL in the cell significantly decreases the dynamic BRET range between $FZD_5$-Nluc and LRP6-Venus upon WNT-3A stimulation, independent of DIX-dependent DVL polymerization and the DVL paralog. We interpret this data as evidence for a role of DVL in orchestrating the arrangement of $FZD_5$-Nluc and LRP6-Venus C-termini in a conformation disfavoring BRET (illustrated in Fig. 2F).

FZD surface expression is subjected to regulation by the ubiquitin ligases ZNRF3 and RNF43 and the secreted protein R-spondin 1, which potentiates WNT signaling by inactivation of these ubiquitin ligases[1,3]. To investigate whether ubiquitination of FZDs affects the outcome of BRET assays between $FZD_5$ and LRP6, we treated HEK293 cells with recombinant R-spondin 1 for two hours prior to WNT-3A stimulation and detected no difference in the respective traces (Supp. Figure 5A). The main ubiquitin ligase regulating $FZD_5$ surface levels is RNF43[30,31]. We co-expressed wt RNF43 or ligase-impaired mutants (R286W, D300G)[32] alongside the BRET biosensors and found that while RNF43 mutants had minimal impact, overexpression of wt RNF43 led to a rapid decline in WNT-3A-induced ΔBRET signals, returning to baseline after a brief initial peak (Supp. Figure 5B). This suggests that endogenous RNF43 does not interfere with BRET assays using over-expressed $FZD_5$-Nluc, whereas overexpressed wt RNF43 significantly reduced WNT-3A-induced ΔBRET, presumably by internalization or degradation of $FZD_5$-Nluc. Notably, R-spondin 1 treatment failed to rescue the phenotype induced by wt RNF43 overexpression in BRET assays (Supp. Figure 5C). We conclude that endogenous ubiquitin ligase activity does not affect the outcome of our BRET assays.

## WNTs can stimulate FZD-LRP6 association without stabilizing β-catenin

To investigate FZD- and WNT-paralog selectivity, we transfected HEK293 cells with representative FZD-Nluc paralogs of the FZD homology clusters that are known to activate WNT/β-catenin signaling ($FZD_{4/5/7}$-Nluc)[33,34] and LRP6-Venus, and stimulated them with diverse recombinant WNTs (WNT-3A, WNT-5A, WNT-10B, and WNT-16B) (Fig. 3A). $FZD_{4/5/7}$ displayed a robust BRET increase when stimulated with WNT-3A with differences in $\Delta BRET_{max}$ ($FZD_5 > FZD_{4/7}$). When stimulated with WNT-5A, most BRET traces showed a transient increase

upon ligand addition, followed by a decrease to baseline. The signal remained slightly above the baseline for $FZD_5$ but displayed a markedly lower $\Delta BRET_{max}$ value compared to WNT-3A stimulation. WNT-10B stimulation resulted in barely detectable ΔBRET signals for $FZD_{4/5/7}$. WNT-16B stimulation resulted in a monophasic ΔBRET increase to a plateau similar to that of WNT-3A stimulation for $FZD_5$, but we observed markedly lower ΔBRET values with $FZD_4$ and particularly $FZD_7$. ($\Delta BRET_{max}$ values $FZD_5 > FZD_4 > FZD_7$).

To determine whether the previously tested WNTs could activate WNT/β-catenin signaling, we investigated phosphorylation of LRP6 and DVL2 upon ligand stimulation in HEK293 cells by Western blotting (Fig. 3E-G). LRP6 phosphorylation was exclusively detected upon stimulation with WNT-3A or WNT surrogate, and not in response to WNT-5A, WNT-10B, or WNT-16B. Only WNT-3A, WNT-5A, and WNT-16B, however, led to a significant upward electrophoretic mobility shift of DVL2 indicative of WNT-induced DVL2 hyperphosphorylation (PS-DVL2)[35]. In an orthogonal approach, we employed a TCF/LEF reporter gene assay (TOPFlash) as a measure of agonist-induced, β-catenin-dependent gene transcription[36,37], yielding essentially the same results in HEK293 cells and HEK293 $\Delta FZD_{1-10}$ cells specifically expressing $FZD_{4/5/7}$ (Fig. 3H; notably, the employed WNT surrogate does not bind to the $FZD_4$-CRD). Other FZD paralogs, whose activation feeds into WNT/β-catenin signaling ($FZD_{1,2,8,10}$), displayed the same activation pattern (Supp. Figure 6A). Of note, WNT-10B, a known activator of WNT/β-catenin signaling[38], induced no reporter gene expression and no LRP6 phosphorylation, but also no DVL2 electrophoretic mobility shift indicative for WNT-induced FZD activation. Therefore we cannot detect signaling activity of WNT-10B in the employed assay formats despite a detectable efficacy in unrelated assay paradigms[39,40]. WNT-16B was of particular interest, as it induced association between $FZD_{4/5/7}$-Nluc and LRP6-Venus but neither LRP6 phosphorylation nor reporter gene activity. To exclude that quantitative FZD degradation upon WNT-16B treatment masks a potential TOPFlash signal, we treated samples with R-spondin 1, which prevents FZD ubiquitination/internalization and dramatically amplifies WNT efficacy[1,3]. Co-treatment of HEK293 cells with WNT and R-spondin 1 greatly amplified the WNT-3A signal, but the WNT-16B signal remained indistinguishable from vehicle in TOPFlash reporter gene assays (Fig. 3I). To strengthen the conclusions from our previous experiments, we investigated the effect of WNT-3A and WNT-16B on cellular β-catenin phosphorylation and protein levels. WNT-3A treatment led to a strong decrease of β-catenin phosphorylation (S33/S37/T41) and to a slight overall increase of total β-catenin levels as compared to vehicle-controlled samples, both indicative for activation of WNT/β-catenin signaling. WNT-16B, on the contrary, affected neither phosphorylated nor total β-catenin compared to vehicle treatment (Supp. Figure 6B-E).

Next, we demonstrated that heat-inactivation of WNT-16B abolished the ΔBRET increase between $FZD_5$-Nluc and LRP6-Venus (Supp. Fig. 7A). Furthermore, we set out to show that WNT-16B behaves similarly to WNT-3A in previously used BRET assay paradigms: Like WNT-3A, WNT-16B displayed no altered ΔBRET trace when using a phosphorylation-insensitive LRP6-5A-Venus mutant (Supp. Fig. 7B), and it also displayed an increased ΔBRET in absence of DVL (Supp. Fig. 7C).

In summary, we observed that WNT-16B does not induce β-catenin-dependent signaling in our cell model but leads to a detectable ΔBRET increase between FZD-Nluc and LRP6-Venus probes. Stimulation with WNT-16B is also not accompanied by LRP6 S1490 phosphorylation characteristic for active WNT/β-catenin signaling. Our observations infer that WNT-induced FZD-LRP6 association as measured by BRET is not sufficient to induce β-catenin-stabilization. Despite the inability of WNT-16B to elicit β-catenin-dependent signaling, the positive efficacy of WNT-16B in the BRET assay and the electrophoretic mobility shift of DVL2 emphasize the protein's ability to interact with its receptors and elicit functional downstream events.

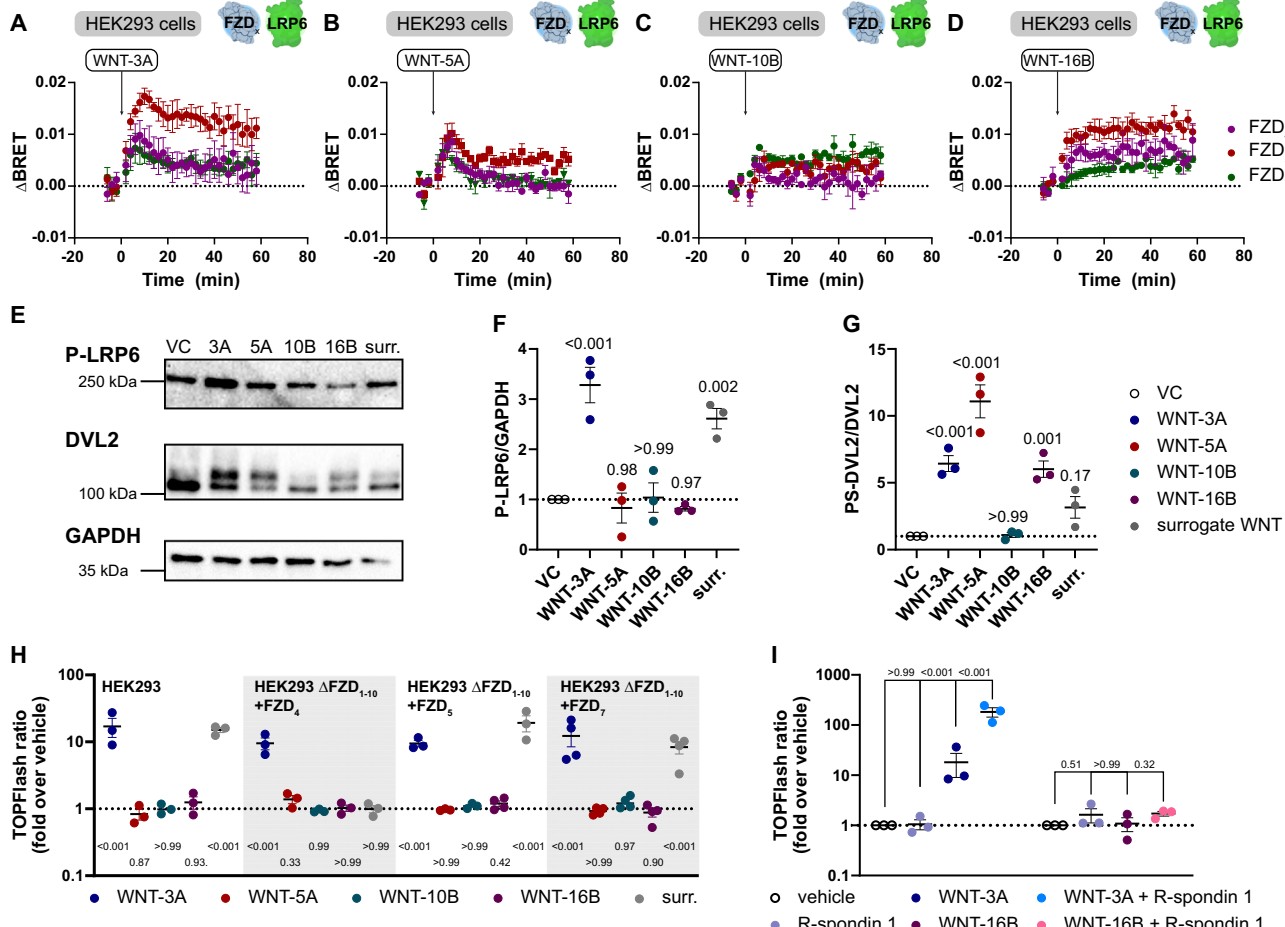

**Fig. 3 | WNTs can stimulate FZD$_x$-LRP6 complex formation in HEK cells without inducing β-catenin stabilization. A–D** Kinetic ΔBRET traces of 1000 ng/ml WNT-3A (**A**), -5A (**B**), -10B (**C**), and -16B (**D**) at FZD$_{4,5,7}$-Nluc (purple, red, and green, respectively) and LRP6-Venus. **E** Immunoblotting of phospho-LRP6, β-catenin, DVL2 and GAPDH (loading control) from whole cell lysates of HEK cells, stimulated for 2 h with 300 ng/ml of WNT-3A, -5A, -10B, and -16B or 1 nM surrogate WNT. The vehicle control (VC) was treated with HBSS and a CHAPS/EDTA mixture corresponding to that present in WNT preparations. **F, G** Densitometric analysis (**F**, ratio of phospho-LRP6 (P-LRP6)/GAPDH; **G**, ratio of phosphorylated and shifted (PS-DVL2/DVL2) of blots shown in E. VC – black circle outlines, WNT-3A – blue, WNT-5A – red, WNT-10B – dark teal, WNT-16B –purple, surrogate WNT – gray. **H.** TOPFlash reporter gene assays in HEK293 and HEK293T ΔFZD$_{1-10}$ cells overexpressing FZD$_{4/5/7}$ stimulated with diverse WNTs (300 ng/ml for 24 h) or WNT surrogate (1 nM for 24 h). The TOPFlash ratio is given as a fold-increase over vehicle control; statistical

analysis was performed with one-way-ANOVA versus vehicle control for each transfection condition. WNT-3A – blue, WNT-5A – red, WNT-10B – dark teal, WNT-16B –purple, surrogate WNT – gray. **I.** TOPFlash reporter gene assays in HEK293 cells stimulated with R-spondin 1 (RSPO1; 100 ng/ml), WNT-3A (300 ng/ml), WNT-16B (300 ng/ml), and WNT-R-spondin 1 combinations for 24 h. Vehicle – black circle outlines, R-spondin 1 – blue-gray, WNT-3A – blue, WNT-16B – purple, WNT-3A + R-spondin 1 – light blue, WNT-16B + R-spondin 1 – pink. Statistical significance was assessed by a one-way ANOVA using Dunnett's post hoc test for multiple comparisons against the vehicle control (reporter gene assay data was $\log_{10}$-transformed prior to statistical analysis). Data points are shown as mean ± SEM of three individual experiments ($n = 4$ in Fid. 2 C, D for FZD$_5$ data and in Fig. 2H for FZD$_7$ TOPFlash data), performed in triplicate in case of BRET assays and reporter gene assays. Exact p-values are detailed in the source data file.

## WNT-3A and WNT-16B induce different transcriptome changes in HEK293 cells

The signaling activity of WNT-16B has remained an enigma for quite some time[41]. For an unbiased assessment of WNT-3A and WNT-16B signaling activity in our model system, we performed poly-A enriched bulk mRNA sequencing of HEK293 cells treated with vehicle control, WNT-3A, or WNT-16B. HEK293 cells were cultured in complete medium supplemented with 10 nM C59 to suppress endogenous WNT secretion for 24 h and were subsequently stimulated for another 24 h with recombinant WNTs. After variance stabilization transformation, the principal component analysis (PCA) of RNA-seq data demonstrated clear separation of cells by treatment (Supp. Fig. 8). Untreated samples clustered quite closely to WNT-3A treated samples, which may hint at basal WNT-3A-like signaling in the sample cells. For differential gene expression analyzes, thresholds were set to p-value < 0.05 and $\log_2$ fold-change of >1.5. Volcano plots (Supp. Fig. 9A-C) depict significantly

up- (red) and down-regulated (blue) transcripts. Stimulation with each WNT resulted in a unique mRNA transcriptome signature relative to vehicle-treated HEK293 cells (Supp. Fig. 9D; see Supp. Dataset 1 for datasets of differentially expressed genes). We subjected subsets of differentially expressed genes (WNT-3A unique, WNT-16B unique, and common) to KEGG pathway and gene ontology (GO) biological processes/cellular component/molecular pathway analyzes, we however identified no significantly ($p_{adj} < 0.05$) regulated pathways in either subset (Supp. Dataset 2). Direct comparison between WNT-3A and WNT-16B treatment also revealed a distinct mRNA signature (Supp. Fig. 9C). Genes regulated by one WNT sometimes correlated with genes regulated by the other WNT (Supp. Fig. 9E), indicating that some transcriptional responses were shared between the treatments, while the majority (1206 out of 1593 comparisons) did not meet significance criteria in both datasets (also see Supp. Fig. 9C). We conclude from this unbiased approach (i) that WNT-16B is transcriptionally

active in HEK293 cells even though it was not inducing a TCF/LEF-dependent response in TOPFlash assays, and (ii) that despite some shared target genes, WNT-3A and WNT-16B mediate largely different transcriptional programs in HEK293 cells, reflecting the observed differences regarding the biophysical readouts and the TOPFlash data.

### WNT-3A and WNT-16B differ in their capacity to induce FZD$_5$-LRP6 clusters

WNT-3A and WNT-16B can interact with the same set of cell surface receptors but mediate different cellular signaling programs with respect to the activation of β-catenin-dependent signaling. How does WNT-receptor activation differ to accomplish distinct cellular responses? We hypothesized that the different signaling outputs of WNT-3A and WNT-16B originate from differences in the molecular interaction between FZD and LRP6, which could not be captured by ensemble methods such as BRET. To this end, we employed dual-color single-molecule fluorescence microscopy to track labeled FZD$_5$ and LRP6 in live cells in response to stimulation with either WNT-3A or WNT-16B.

We transiently transfected N-terminally tagged SNAP-FZD$_5$ and Halo-LRP6 constructs into Chinese Hamster Ovary K1 (CHO-K1) cells to achieve low near-physiological molecule density (FZD$_5$: 0.42 ± 0.1, LRP6: 0.59 ± 0.15 molecules/μm$^2$). Receptors were labeled with saturating concentrations of SNAP SiR-647 and Halo R110 fluorophores, respectively, and imaged with fast, multi-color total internal reflection fluorescence (TIRF) microscopy, combined with single-particle tracking (Fig. 4A, B). Data were acquired both under basal (Supp. Video 1+2 for raw movies and single-particle tracking with additional plotting of interactions) and after early (2 – 10 min) and late (11 – 25 min) stimulations with WNTs (Supp. Video 3-10). When analyzed by time-averaged mean squared displacement, FZD$_5$ and LRP6 molecules explored a range of diffusion profiles on the plasma membrane, alternating between confined and random Brownian diffusion (Supp. Fig. 10A). Both WNT-3A and WNT-16B stimulations increased the proportion of confined FZD$_5$ and LRP6 molecules (Fig. 4C). To estimate the frequency and duration of FZD$_5$ and LRP6 interactions, we applied previously developed methods based on deconvolution of apparent colocalization times with those of random colocalizations[42]. Random colocalization times were estimated by imaging SNAP-β$_2$-adrenoceptors (β$_2$ARs), a prototypical GPCR with similar diffusion properties to that of FZD$_5$ (expressed at similar densities 0.4 ± 0.08 molecules/μm$^2$), and Halo-LRP6. In the absence of WNTs, FZD$_5$ and LRP6 molecules did not colocalize for longer than what we measured for random colocalizations between SNAP-β$_2$AR and Halo-LRP6 confirming our data from BRET acceptor titration experiments (Supp. Figure 2A, Supp. Videos 11+12).

Following stimulation, both WNT-3A and WNT-16B caused a substantial increase in FZD$_5$ and LRP6 association rates at both early and late stimulation time-points. Notably, WNT-3A induced a ~1.5 and ~10-fold increase in association rate ($k_{on}$) compared to WNT-16B at early and late stimulation time-points, respectively (Fig. 4D, left). We estimated that following early and late WNT-16B stimulation, FZD$_5$ and LRP6 interactions lasted ~0.76 s and ~1.57 s, respectively ($k_{off}$ early = 1.32 s$^{-1}$, 95% confidence interval (CI): 0.81 - 1.84; $k_{off}$ late = 0.64 s$^{-1}$, 95% CI: 0.52 - 0.75). WNT-3A induced a ~2-fold (early and late) increase in FZD$_5$-LRP6 interaction times, compared to WNT-16B ($k_{off}$ early = 0.65 s$^{-1}$, 95% CI: 0.37 - 0.92; $k_{off}$ late = 0.3 s$^{-1}$, 95% CI: 0.24 - 0.36) (Fig. 4D, right). Importantly, WNT-3A interaction times at late stimulation time-points were reaching the limit of the observation window in our experiments, which indicated that the true length of interactions following WNT-3A late stimulation are in fact longer than measured. A spatial confinement analysis revealed that both ligands induced receptor interactions in a co-confined state, where FZD$_5$ and LRP6 are confined together, rather than a co-diffusing state, where both receptors move together across the plasma membrane. However, only WNT-3A showed statistically significant increase in receptor co-confinement. This shift from co-diffusion to co-confinement is reflected by the inverted pattern in the graph depicting the relative fraction of detected interactions (Fig. 4E). In addition, at the late stimulation time-point, we observed that WNT-3A induced aggregation of both FZD$_5$ and LRP6 into clustered, higher-order complexes, which were not observed when cells were stimulated with WNT-16B (Fig. 4F). Furthermore, we exploited the single-molecule data to analyze the size of such complexes. Monomeric, fluorescent particles are expected to photobleach in a single step, whereas higher-order oligomers follow a stepwise photobleaching pattern (Supp. Fig. 10B). In response to WNT-16B-stimulation, stepwise photobleaching analysis showed that co-confined FZD$_5$ and LRP6 molecules were largely monomeric (~60% and 80%, respectively). Conversely, WNT-3A-stimulated cells had a smaller fraction of monomeric co-confined FZD$_5$ and LRP6 molecules (~20%) and higher numbers of molecules that photobleached in multiple steps (up to 15), indicating the presence of higher-order clusters (Fig. 4G). Notably, by the time receptors had aggregated into higher-order clusters in the single-molecule fluorescence microscopy experiments, ΔBRET traces had already reached their plateau (>10 min).

While stimulation with both WNTs increased receptor confinement, co-confinement, and interaction duration, the characteristic formation of FZD$_5$-LRP6 higher-order clusters was more prominent with WNT-3A than WNT-16B. These findings provide a mechanistic explanation for the lack of effective WNT/β-catenin signaling upon WNT-16B stimulation. Thus, ligand-induced receptor association and confinement, as seen with WNT-16B, is not sufficient for WNT/β-catenin signaling, which requires the formation of higher-order clusters.

### LRP6 clusters independent of its phosphorylation state

We made use of the phosphorylation-insensitive LRP6-5A mutant in single-molecule tracking experiments to explore whether the formation of FZD$_5$-LRP6 clusters is either a prerequisite for or a consequence of LRP6 phosphorylation, shedding light on the sequence of events in the initiation of WNT/β-catenin signaling. To this end, we stimulated CHO cells expressing SNAP-FZD$_5$ and Halo-LRP6-5A with WNT-3A as detailed above. We measured an increased proportion of confinement for both SNAP-FZD$_5$ and Halo-LRP6-5A (Fig. 5A) in early and late stimulation movies, observing the formation of larger FZD$_5$-LRP6-5A clusters as described for wt LRP6 (compare Fig. 5B and Fig. 5F). WNT-3A stimulation of FZD$_5$ and LRP6-5A similarly resulted in an increased association rate between both receptors, which further increased after prolonged stimulation. We estimated that interactions between FZD$_5$ and LRP6-5A lasted for ~1.6 s and ~2.0 s seconds after early and late stimulation with WNT-3A, respectively (Fig. 5C). Moreover, we observed that, similar to experiments performed with wt LRP6, both receptors interacted in a co-confined state rather than in a co-diffusing state (Fig. 5D). Finally, stepwise photobleaching analysis confirmed that only ~26% and ~33% of FZD$_5$ and LRP6-5A molecules at the cell surface, respectively, interacted in a monomeric state, while the vast majority of both clustered in groups of 2-8 specimens of each receptor (Fig. 5E, F).

The WNT-3A-induced interactions between FZD$_5$ and LRP6 and LRP6-5A behave clearly similar with respect to most analyzes we employed, however, we observed two differences: Firstly, large clusters > 8 specimens of each receptor per cluster were virtually not detected in cells expressing LRP6-5A, but made up roughly 9% of clusters in cells expressing wt LRP6. Secondly, the $k_{on}$ observed in late-stage WNT-3A stimulation was ~3-fold lower for LRP6-5A (0.25 μm$^2$ molecule$^{-1}$ s$^{-1}$) than it was for wt LRP6 (0.72 μm$^2$ molecule$^{-1}$ s$^{-1}$; $p < 0.001$, Mann-Whitey nonparametric test). However, our data demonstrate that WNT-3A-induced clustering of LRP6 and FZD$_5$ is

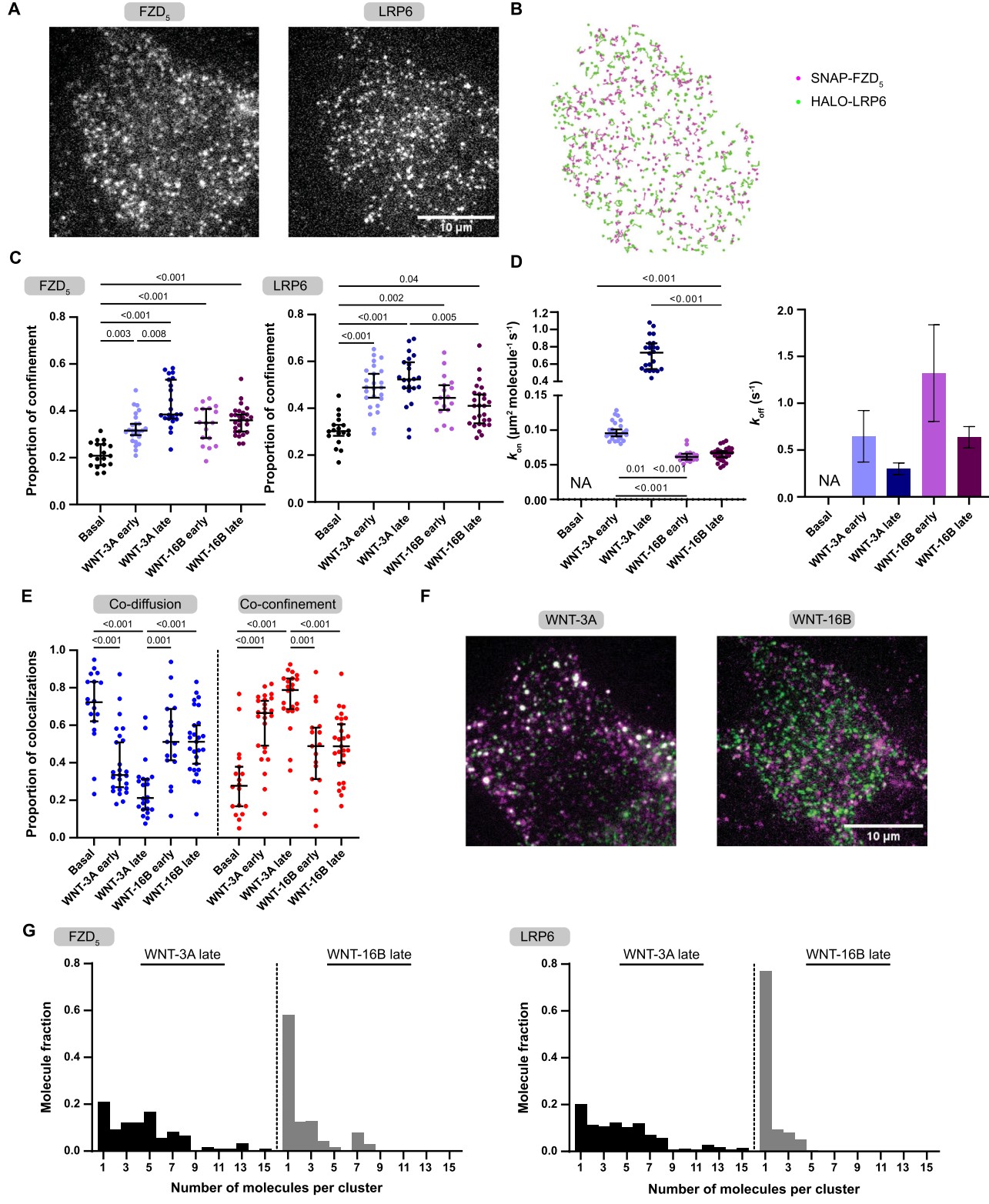

not a consequence of phosphorylation of LRP6 and is maintained with the mutant LRP6-5A.

## Discussion

In the present work, we used a combination of BRET assays, biochemical signaling readouts, transcriptomics, and single-molecule fluorescence microscopy to further delineate the signal initiation and specification in WNT/β-catenin signaling. We primarily base our work on a pharmacological comparison between WNT-3A and WNT-16B (see

Fig. 6A), acting on $FZD_5$ and LRP6, which serve as model receptors throughout the study. The central finding of this work is that the addition of WNT-16B leads to association of $FZD_5$ and LRP6 but does not feed into WNT/β-catenin signaling. WNT-16B is an endogenous ligand that challenges the tenet that β-catenin signaling necessarily follows from ligand-induced interaction between FZDs and LRP6, as implied by the development of WNT surrogates that act as extracellular FZD-LRP6 crosslinkers to efficaciously activate WNT/β-catenin signaling[9-11,43,44].

**Fig. 4 | WNT stimulation increases interactions between FZD₅ and LRP6 and confines the membrane receptors. A**. Representative single-molecule images showing SNAP-FZD₅ molecules, labeled with SNAP SiR-647 (left) and Halo-LRP6 molecules, labeled with Halo R110 (right). **B**. Single-molecule trajectory traces of SNAP-FZD₅ (magenta) and Halo-LRP6 (green). **C**. Proportion of molecular confinement at basal and following 100 nM WNT-3A and 100 nM WNT-16B stimulations. Basal – black, WNT-3A early – light blue, WNT-3A late – blue, WNT-16B early – light purple, WNT-16B late – purple. **D**. Estimated $k_{on}$ (left) and $k_{off}$ (right) values of FZD₅-LRP6 interactions at basal and following 100 nM WNT-3A and 100 nM WNT-16B stimulations. Basal – black, WNT-3A early – light blue, WNT-3A late – blue, WNT-16B early – light purple, WNT-16B late – purple. **E**. Distributions of co-diffusion (blue) and co-confinement (red) events at basal and 100 nM WNT-3A and 100 nM

WNT-16B stimulated conditions. **F**. Representative, dual-color single-molecule images showing SNAP-FZD₅ molecules, labeled with SNAP SiR-647 (magenta) and Halo-LRP6 molecules, labeled with Halo R110 (green) following 100 nM WNT-3A and 100 nM WNT-16B stimulations at late time-point. **G**. Cluster analysis showing the distributions of photobleaching steps following 100 nM WNT-3A and 100 nM WNT-16B stimulations at late time-point. (C-E) Data points are shown as median ± 95% confidence interval. Early stimulation: 2-10 min, late stimulation: 11-25 min. Statistical comparisons were made by Kruskal-Wallis followed by Dunn's multiple comparison test. $n = 18, 25, 22, 17, 27, 25$ cells for FZD₅-LRP6 basal, WNT-3A early, WNT-3A late, WNT-16B early, WNT-16B late, and β₂AR-LRP6, respectively, from six independent experiments. See also Supp. Movies 1-12. Exact p-values are detailed in the source data file.

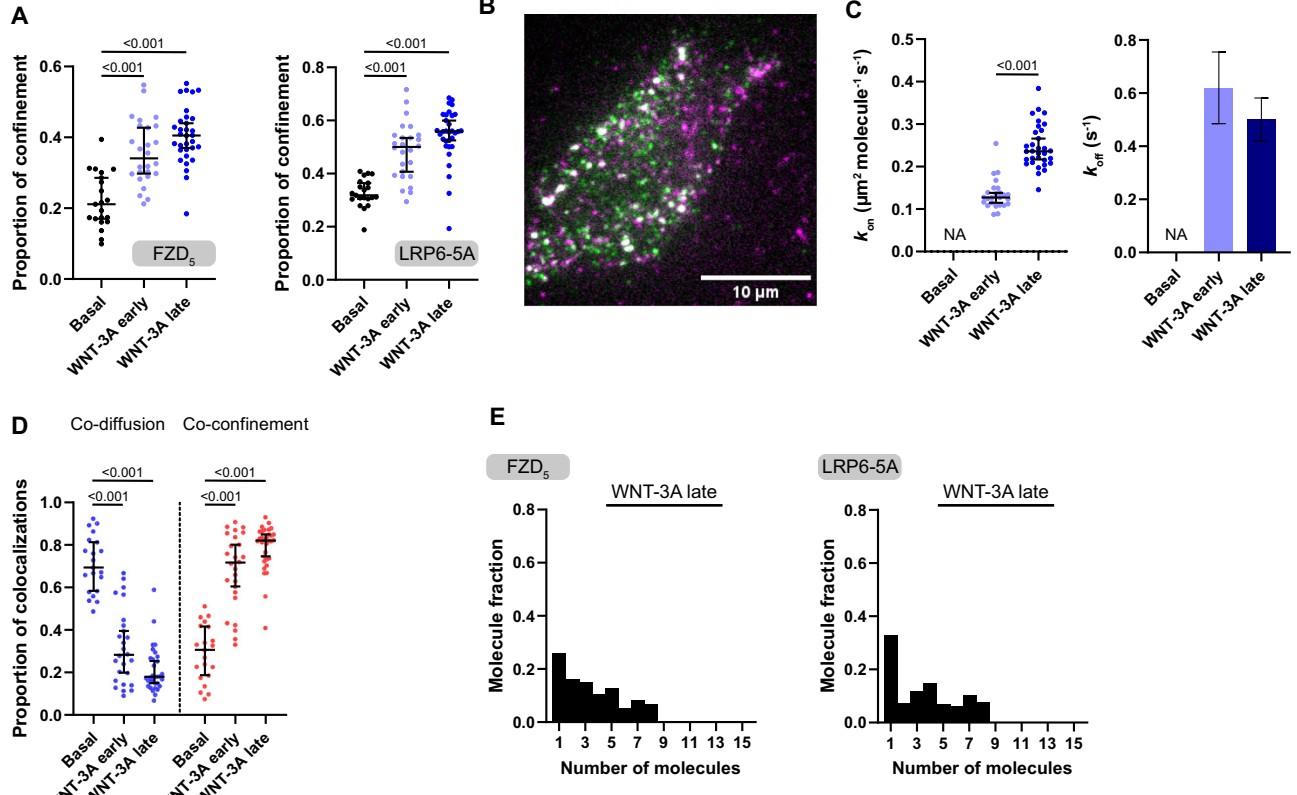

**Fig. 5 | WNT-3A stimulation increases interactions between FZD₅ and the phospho-insensitive LRP6-5A mutant and confines the receptors. A** Proportion of SNAP-FZD₅ and Halo-LRP6-5A molecular confinement at basal (black) and following 100 nM WNT-3A (early – light blue, late – dark blue) stimulation. **B** Representative, dual-color single-molecule image showing SNAP-FZD₅ molecules, labeled with SNAP SiR-647 (magenta) and Halo-LRP6-5A molecules, labeled with Halo R110 (green) following 100 nM WNT-3A stimulation at late time-point. **C**. Estimated kon (left) and koff (right) values of FZD₅-LRP6-5A interactions at basal and following 100 nM WNT-3A stimulation (early – light blue, late – dark blue).

**D** Distributions of co-diffusion (blue) and co-confinement (red) events at basal and 100 nM WNT-3A stimulated conditions. **E** Cluster analysis showing the distributions of photobleaching steps following 100 nM WNT-3A stimulation at late time-point. **A**, **C**, **D** Data points are shown as median ± 95% confidence interval. Early stimulation: 2-10 min, late stimulation: 11-25 min. Statistical comparisons were made by Kruskal-Wallis followed by Dunn's multiple comparison test (**A**, **D**) and Mann-Whitney test **C**. $n = 21, 26, 32$ cells for FZD₅-LRP6-5A basal, WNT-3A early, WNT-3A late, respectively, from three independent experiments. See also Supp. Movies 13–18. Exact p-values are detailed in the source data file.

The established direct NanoBRET assay between FZD₅ and LRP6 provides insight into the process of ligand-induced association between FZDs and LRP6. Firstly, we observed that FZD₅-LRP6 association is independent from LRP6 phosphorylation, inferring that receptor association is not modulated by downstream signaling events (see Fig. 1G+H, Supp. Figure 3). Secondly, the interaction between FZD₅ and LRP6 is exclusively ligand-induced. While this opposes a study claiming direct and constitutive interaction between FZD₈/LRP6 extracellular domains[23], it is well in line with observations from single-molecule fluorescence microscopy[9].

Furthermore, we found that the orientation of the FZD₅ and LRP6 C-termini is allosterically modulated by DVL binding to FZD,

where the presence of DVL results in a lowered WNT-induced ΔBRET_max (see Fig. 2, Supp. Figure 4, Supp. Fig. 7C). This observation was independent of DVL polymerization. We speculate that DVL binding to FZD₅ may induce a conformation of the FZD₅ C-terminus which is not optimal for Nluc-Venus energy transfer. We exclude that our observations originate from different receptor surface expression levels (Supp. Figure 4A, B). DVL generally appears to be dispensable for the initial ligand-induced association of FZD and LRP6, yet the increased dynamic BRET range provides evidence that endogenous DVL already interacts with the FZD-LRP6 complex at this early stage of signal initiation. Combined with the recent observation that a WNT-induced change in the FZD-DEP interface was

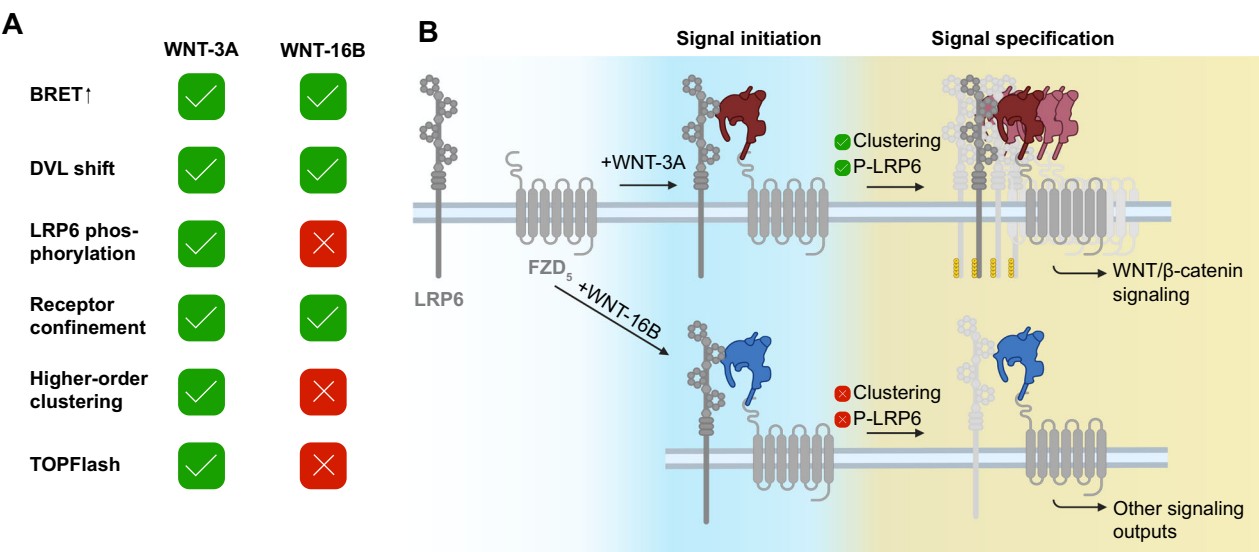

**Fig. 6 | A proposed two-step model for the initiation of signaling by WNT-3A and WNT-16B. A** Direct comparison of the effects of WNT-3A and WNT-16B as probed in this study. **B** Two-step model of signal initiation and specification in WNT/β-catenin signaling. In a first step, WNT binding leads to ligand-induced association of FZD₅ and LRP6 (signal initiation) for both WNT-3A (top, red) and WNT−16B (bottom, blue). Upon higher-order receptor clustering and LRP6 phosphorylation, the signal is specified into WNT/β-catenin signaling (top, red). If these hallmarks are not met, the FZD-WNT complex can signal via other signaling pathways (bottom, blue). It is unclear whether LRP6 is involved in signal specification in that case. Created in BioRender. Voss, J. (2025) https://BioRender.com/e80oa99. Parts of this figure were created with BioRender.com released under a Creative Commons Attribution-NonCommercial-NoDrivs 4.0 International license.

independent of LRP5/6[17,45], our data suggest that FZD-DVL interactions precede LRP6 association.

The observed ligand-induced association between FZDs and LRP6 as measured by BRET occurs in response to WNTs that do not elicit a TOPFlash signal (i.e., several WNT-FZD-LRP6 combinations result in a positive BRET shift, but not in a reporter gene signal, compare Fig. 3A-D and Fig. 3H). Thus, our experiments dissect receptor association from the WNT/β-catenin signaling output assessed by TOPFlash. Therefore, we suggest that BRET measurements do not exclusively capture the formation of signaling-competent receptor clusters, but especially the preceding FZD-LRP6 association and (co)-confinement (see also Fig. 4C-E), which does not necessarily provide a signaling-competent platform. The signal specification into WNT/β-catenin signaling must be taking place downstream of the initial association of FZD₅ and LRP6. Of note WNT-5A, a WNT that generally does not act via the WNT/β-catenin pathway[46,47], resulted in a low positive ΔBRET at FZD₅. This aligns well with a report showing physical interaction between LRP6 and WNT-5A[48], but challenges the hypothesis that WNT signal specification is solely achieved by engagement of different co-receptors[49,50].

RNA sequencing studies confirmed that both WNT-3A and WNT-16B induce changes in the transcriptome of HEK293 cells and are therefore indubitably active. However, as all RNA sequencing experiments were conducted in HEK293 cells, it is unclear whether differentially expressed genes are reflective of WNT target genes in physiological contexts, such as stem cell niches or developing tissues. An RNA sequencing study of WNT-16B signaling in limbal epithelial stem cells identified cytokine-cytokine receptor interactions as a highly enriched signaling pathway, which we did not identify in HEK293 cells[51]. Additionally, well-described WNT/β-catenin target genes such as *AXIN2*, *NOTUM*, and *LGR5* were not upregulated in our samples, even though we observed hallmarks of β-catenin signaling by Western blotting (Fig. 3E-G, Supp. Figure 6B-E). We measured elevated basal levels of phosphorylated LRP6 (Fig. 3E, F) in HEK293 cells and detected minor transcriptomic changes in response to WNT-3A compared to WNT-16B (Supp. Fig. 8, Supp. Fig 9A, C), perhaps hinting at basal WNT/β-catenin signaling. We therefore caution against interpreting gene expression changes in HEK293 cells as blueprint of WNT signaling activity in physiological contexts.

Using single-molecule fluorescence microscopy we established that at comparable receptor expression levels, WNT-3A and WNT-16B behave differently with regards to the extent of receptor clustering over time (Fig. 4G), yet both ligands still confine FZD₅ and LRP6 (Fig. 4C, E), underlining their ability to interact with both receptors. It is of paramount importance to emphasize that WNT-16B stimulation still translated into substantial transcriptomic changes in HEK293 cells, despite being inactive in TOPFlash reporter gene assays (compare Fig. 3H+I with Supp. Fig. 9B, C), suggesting the activation of β-catenin-independent pathways. Whether LRP6 is relevant for the signal specification after WNT-16B stimulation or whether the signal is further specified by co-receptors other than LRP6 (or even in a DVL-independent manner) cannot be stated from results obtained here, and a further dissection of WNT-16B signaling mechanisms in HEK293 cells is beyond the scope of this work.

We observed higher-order clustering of receptors only after approx. 10 min of WNT-3A stimulation. Yet, receptor confinement and co-confinement had been observed prior to that, indicating that lower-order clustering or association of FZD₅ and LRP6 precede the formation of larger, signaling-competent signalosomes feeding into the WNT/β-catenin pathway. As a result, the term WNT/β-catenin signalosome describing a signaling-competent receptor agglomerate must be defined as a higher-order cluster rather than the minimal WNT/FZD/LRP/DVL complex as often depicted schematically[3]. From a kinetic perspective, BRET traces reached their plateau within 10 min, again suggesting that they capture initial FZD-LRP5/6 co-confinement upon WNT stimulation and not higher-order clustering. The later data points obviously monitor both association and clustering, however the formation of higher-order structures does not further elevate the BRET signal.

Lastly, single-molecule tracking experiments with the non-phosphorylatable LRP6-5A mutant strongly suggest that higher-order clustering of LRP6 is largely independent from its phosphorylation (Fig. 5), a question which was previously not addressed with direct evidence. We observed minor differences in FZD-LRP6-5A cluster sizes

and receptor association rate when compared to wt LRP6, therefore we cannot exclude that LRP6 phosphorylation further contributes to FZD₅-LRP6 clustering (Fig. 5C, E). Our data indicate, in contrast to recent suggestions[52], that LRP6 clusters with FZDs in response to WNT-3A stimulation independent from LRP6 phosphorylation. Subsequent binding of casein kinases and glycogen synthase kinase 3 mediating phosphorylation of the LRP6 C-terminus could then contribute to specify efficient WNT/β-catenin signaling.

How does signal initiation elicited by WNT-3A acting at FZD₅ and LRP6 differ from that in response to WNT-16B acting at the same receptors? We demonstrate that further mechanisms beyond ligand-induced FZD-LRP6 association are required to activate the WNT/β-catenin signaling pathway – specifically the formation of higher-order FZD₅-LRP6 clusters and the subsequent phosphorylation of LRP6 are absent upon WNT-16B stimulation. These additional mechanisms are also activated by WNT surrogates, that fulfill all hallmarks of WNT/β-catenin signaling, even when engaging FZD₆, a receptor that typically does not signal via β-catenin[43]. Multivalency of WNT surrogates was described to significantly boost their efficacy, and one study found that a monovalent crosslinker (simultaneously binding one FZD and one LRP6 molecule) was inactive in reporter gene assays[11]. Yet WNTs themselves and first-generation surrogates supposedly engage FZDs in a 1:1 stoichiometry[47,53], which is in accordance with our stepwise photobleaching analysis suggesting a similar average amount of both FZD₅ and LRP6 molecules in each cluster. Therefore, it is currently unclear which molecular properties of a ligand are decisive for whether a FZD-WNT-LRP6 complex passes the checkpoints to initiate WNT/β-catenin signaling, or not. We have previously shown that WNT-16B stabilizes a different conformation than WNT-3A on the intracellular side of FZD₅[40], suggesting that it could form a structurally distinct effector binding site. Yet, WNT-3A and WNT-16B show a rather similar dynamic effect on the orientation of the DVL2 DEP domain relative to FZD₅[45]. There are, however, notable reports of WNT-16B/β-catenin signaling: Mouse WNT-16 was described to have a weak agonistic efficacy in WNT/β-catenin signaling in MC3T3 mouse osteocyte precursor cells[54], and WNT-16B promoted chemotherapy resistance via activation of β-catenin signaling in human prostate cancer cells[55]. Speculatively, cell type-specific expression of a co-factor absent in HEK293/CHO cells might be required to initiate WNT/β-catenin signaling upon WNT-16B stimulation. Additionally, β-catenin-independent functions of WNT/LRP6 signaling have been described, such as WNT-dependent stabilization of proteins and WNT/TOR signaling[56,57]. Furthermore, WNT-16B did not induce TOPFlash via any of the FZD paralogs known to activate the WNT/β-catenin pathway (Fig. 3H, Supp. Figure 6A). Therefore, we do not consider FZD paralog-selective activation of WNT/β-catenin signaling by WNT-16B as likely[38,56,57].

In conclusion, we propose a two-step model for the efficient activation of WNT/β-catenin signaling, in which WNT stimulation initially leads to quick association and co-confinement of FZDs and LRP6 (signal initiation). In a second step, the signaling outcome of the respective ligand is specified in a checkpoint-like manner (Fig. 6B): If the complex shows hallmarks of WNT/β-catenin signaling such as higher-order receptor clustering and, subsequently, LRP6 phosphorylation, it will activate WNT/β-catenin signaling (Fig. 6B red WNT, top). Thus, our data allow us to reformulate the concept of signalosome formation putting LRP6 phosphorylation downstream of higher-order FZD-LRP6 cluster formation. Furthermore, WNT stimulation can trigger transcriptional changes irrespective of formation of higher-order clusters, presumably through other pathways (Fig. 6B blue WNT, bottom). The molecular properties required for a FZD-WNT-LRP6 complex to feed into WNT/β-catenin signaling remain unclear and warrant further investigation. This model underscores that ligand-induced association of FZD and LRP5/6 is necessary but not sufficient for WNT/β-catenin signaling but also provides a framework for understanding why diverse WNTs, despite engaging the same receptors, result in distinct signaling outcomes.

## Methods

### Plasmids

Constructs for recombinant protein expression used in this study are listed in Table S1. All plasmids encode human receptor genes. Constructs generated in this study were created by standard seamless cloning techniques following the manufacturer's protocol (New England Biolabs #E2611)[58]. This entails all LRP5/6 constructs and FZD-Venus constructs. The template sequences for LRP5 and LRP6 were obtained from Addgene (#115907 and #27282, respectively)[8,26]. FZD constructs generally followed the following architecture: 5-HT₃ₐ receptor signal peptide (MRLCIPQVL-LALFLSMLTGPGEGSR) – HA-Tag – BamH1 restriction site – FZD coding sequence (no signal peptide) – 10 residue linker – Nluc/mVenus. LRP5/6 constructs were constructed as follows: Influenza hemagglutinin signal peptide (MKTIIALSYIFCLVFA) – FLAG-tag – LRP5/6 coding sequence (no signal peptide) – QPVAT-linker – Nluc/mVenus. FZD₅ and LRP6 constructs used for single-molecule fluorescence microscopy followed the design 5-HT₃ₐ receptor signal peptide – BamH1 site (GS) – coding sequence, without further tags. Mutations, as described for DVL2, were introduced by site-directed mutagenesis (GeneArt™ Site-Directed Mutagenesis System, Thermo Fisher #A13282). LRP6-5A mutants were created with help of a gBlock (IDT) of the mutant LRP6 C-tail, which was cloned into linearized LRP6-Nluc/Venus lacking the respective part by Gibson assembly.

### Cell culture and transient transfection

HEK293 cell lines were cultured in Dulbecco's Modified Eagle Medium (DMEM; Gibco, Thermo Fisher Scientific, Waltham, MA), containing 10% FCS (Gibco), penicillin (100 U/ml, Gibco), and streptomycin (100 μg/ml, Gibco) at 37 °C in a humidified atmosphere containing 5% $CO_2$. CHO-K1 cells were cultured in DMEM/F12 medium under the same conditions. When reaching approx. 80% confluency, cells were passaged by trypsinization. Cells were routinely confirmed to be free of mycoplasma contamination by a strip test kit (InVivoGen, Toulouse, France). HEK293 cells were obtained from Thermo Fisher Scientific (female origin, #R70507). HEK293 ΔLRP5/6 and HEK293 ΔFZD₁₋₁₀ cells were a gift from B. Vanhollebeke (Université Libre de Bruxelles, Belgium). HEK293 ΔDVL1-3 cells were a gift from M. Gammons (Cambridge Institute for Medical Research, UK).

For transient transfections, cells were trypsinized and the cell number was adjusted to 350,000/ml with complete medium. Each ml of cell suspension was transfected with 1 μg of cDNA, and all cDNA ratios will be given in % of the total amount of cDNA in the mixture. If the biosensor/protein of interest-encoding cDNA(s) did not add up to 100%, an empty pcDNA3.1 vector was used to adjust the cDNA amount. Transient transfection was performed using PEI MAX® (Polysciences, Warrington, PA) in a three-fold weight excess relative to DNA. The PEI-DNA mixture was incubated in a total volume of 100 μl DPBS per μg DNA for 15-45 min before addition of cells. The transfected cells were then directly added to the respective assay plate and incubated as detailed in the respective sections.

For single-molecule experiments, CHO-K1 cells were seeded onto ultraclean 25 mm round glass coverslips at a density of $3 \times 10^5$ cells per well in a 6-well plate. On the next day, they were transfected with plasmids encoding SNAP-FZD₅ or a SNAP-β₂ adrenoceptor and Halo-LRP6/LRP6-5A using Lipofectamine 2000, following the manufacturer's protocol. Cells were labeled and imaged by single-molecule microscopy 3 hours after transfection to obtain low close-to physiological expression levels[42,59].

## Recombinant proteins

Recombinant WNTs and R-spondin 1 were purchased from Biotechne with the following catalog numbers: human WNT-3A (5036-WN-010), human/mouse WNT-5A (645-WN-010/CF), recombinant human WNT-10B (7196-WN-010), human WNT-16B (7790-WN-025), R-spondin 1 (4645-RS-025). To heat-inactivate WNT-16B, it was subjected to 2 cycles of 60 °C and -20 °C for 20 min each. Vehicle controls corrected for CHAPS, EDTA and BSA present in protein preparations WNT surrogate was purchased from U-protein express/IPA Therapeutics (WNT Surrogate-Fc Fusion Protein N001-0.5 mg).

## Enzyme-linked immunosorbent assay (ELISA)

Transfected cells were seeded into a poly-D-lysine-coated clear 96 well plate at a density of 35,000 cells per well and incubated for 40 h. The medium was aspirated, and cells were incubated in primary antibody solution (α-FLAG-M2, Sigma, F1804, or α-HA, abcam, ab9110, Lot No. GR3425636-2; both 1:1000) in ELISA Buffer (DPBS supplemented with $MgCl_2$ and $CaCl_2$ (Gibco) + 1% bovine serum albumin (BSA)) for 1 h at 4 °C. The cells were washed five times for 5 min with ice-cold washing buffer (DPBS supplemented with $MgCl_2$ and $CaCl_2$ (Gibco) + 0.5% BSA) and then incubated with secondary HRP-conjugated anti-mouse (Thermo Fisher Scientific, 31430) or anti-rabbit (Thermo Fisher Scientific, 31460) secondary antibody for 1 h at 4 °C (1:2500 in ELISA buffer), followed by five times washing performed as above. After the last washing step, 50 μl of 3,3′,5,5′-tetramethylbenzidine (Sigma Aldrich, T8665) were added to the cells and the plate was incubated in the dark for 30 min at room temperature. Then, the wells were acidified with 50 μl of 2 M HCl and the absorption was measured at 450 nm at a TECAN Spark multimode plate reader.

## Bioluminescence resonance energy transfer assays

Cells were transfected with 1% of the indicated Nluc donor plasmid and 10 or 30% of the Venus-tagged BRET acceptor for FZDs and LRPs, respectively, as well as 20% of MESD. Experiments with co-expression of DVL2 or RNF43 were performed with 10% of the corresponding plasmid. For BRET assays, 35,000 transfected cells were seeded into white 96-well plates and incubated for 40-48 h before measurement. After washing the cells once with HBSS, 80 μl of HBSS + 0.1% BSA was added to each well. Luciferase substrate (10 μl furimazine (Promega), 1:100 in assay buffer) was added to each well 6 min before measurement. For kinetic reads, basal Nluc and Venus signal was read 3 times with a 2 min interval before ligand stimulation. After ligand addition, signals were read for another 60 min in 2 min intervals. Data were double baseline-corrected for the baseline and for vehicle control. For acceptor titration experiments, fluorescence was read three times prior to substrate addition to determine the acceptor expression levels (excitation 485/20 nm; emission 535/25 nm), and 6 min after substrate addition, luminescence and acceptor fluorescence were read again thrice to measure the BRET ratio. Data were baseline corrected for background fluorescence and background BRET (at 0% acceptor expression) to obtain net BRET values. All measurements were performed with a Spark multimode plate reader (Tecan) at a temperature of 37 °C. Nluc donor emission was detected at wavelengths of 445-485 nm, Venus acceptor emission was detected at 520−560 nm, each with an integration time of 100 ms. When R-spondin 1 was present in BRET experiments, it was pre-incubated for 2 h at a concentration of 100 ng/ml before the addition of WNTs.

## Western blotting

Samples for Western Blotting were obtained by seeding 300,000 cells in a 96-well plate for 40-48 h in complete medium supplemented with 10 nM of the porcupine inhibitor C59. If indicated, cells were transfected as described above. Stimulation of the cells was performed 2 h before lysis by addition of 300 ng/ml WNT or 1 nM WNT surrogate.

After the stimulation period, the medium was removed and cells were lysed by addition of 2x Laemmli buffer supplemented with 200 mM dithiothreitol and subsequent sonication. Lysates were heated to 60 °C for 20 min before separation on a 7.5% Mini-Protean TGX precast gel (BioRad) with a constant voltage of 120 V. Protein transfer onto polyvinylidene difluoride membranes was performed with the TransBlot Turbo transfer system (BioRad) and the manufacturer's "Mixed molecular weight" transfer protocol. Membranes were subsequently blocked in blocking buffer (TBS-T (25 mM Tris-HCl, 150 mM NaCl, 0.1% Tween 20, pH 7.6) + 5% dry milk powder) for 1 h and incubated in primary antibody (α-P-LRP6, 1:1000, Cell Signaling Tech., Cat. No. 2568, Lot No. 9; α-DVL2, 1:1000, Cell Signaling Tech., Cat. No. 3216, Lot No. 2; α-GAPDH, 1:2500, Cell Signaling Tech., Cat. No. 2118, Lot No. 61; α-P-β-catenin, 1:1000, Cell Signaling Tech., Cat. No. 9561, Lot No. 5; α-total β-catenin, 1:1000, BD Biosciences, Cat. No. 610154, Lot No. 3242871; all diluted in blocking buffer). Membranes were washed 5×5 min with TBS-T before addition of secondary HRP-coupled anti-rabbit antibody (Thermo Fisher Scientific, 31460; 1:5000 diluted in blocking buffer). After 1 h, membranes were washed again for 5 × 5 min with TBS-T. Blots were developed using Clarity Western ECL substrate (Bio-Rad) following the instructions of the manufacturer in a ChemiDoc chemiluminescence reader (Bio-Rad).

## TOPFlash reporter gene assays

HEK293 cells or HEK293 $\Delta FZD_{1-10}$ cells were transfected with 25% of an 8x SuperTOPFlash reporter plasmid (Addgene #12456), 5% of a pRL-TK control plasmid (Promega), 20% of HiBiT-FZD constructs (if indicated), and LRP6 constructs as annotated (1% if not mentioned otherwise), and were seeded at a density of 35,000 and 45,000 cells per well, respectively, into a PDL-coated white 96-well flat bottom plate. After 16-20 h of incubation at 37 °C/5% $CO_2$ in complete medium, the medium was removed and replaced with starvation medium (regular DMEM without supplements) + 10 nM of the porcupine inhibitor C59. At the same time, cells were stimulated with 300 ng/ml of the indicated WNT or 1 nM of WNT surrogate, and 100 ng/ml R-spondin 1 when indicated. Each stimulation condition was paired with an appropriate vehicle control. Cells were stimulated for 24 h, washed once with HBSS and lysed with 20 μl of 1x passive lysis buffer (Dual Luciferase Assay Kit, Promega, #E1910) for 20 min at room temperature. Then, 20 μl of LARII reagent was added and firefly luminescence was immediately read (550-620 nm, 2 s integration time) using a Spark multimode plate reader (Tecan). Subsequently, 20 μl of Stop-and-Glo reagent spiked with Renilla luciferase substrate were added, and Rluc luminescence was read (445-530 nm, 2 s integration time). The raw TOPFlash ratio was obtained by dividing Fluc counts by Rluc counts, which was afterwards normalized to vehicle control and/or a pcDNA-transfected control to calculate agonist-induced or protein expression-induced TOPFlash ratios.

## Bulk mRNA sequencing

HEK293 cells were cultured in a 6-well plate (500,000 cells per well) in complete medium for h in a total volume of 2 ml. Cells were then stimulated with vehicle control, 300 ng WNT-3A, or 300 ng/ml WNT-16B and 10 nM of the Porcupine inhibitor C59 was added to the cells. Medium was aspired and the cells were trypsinated, centrifuged, and put on ice. was extracted from the cell suspension using the RNeasy Mini Kit (Qiagen). Subsequent mRNA library preparation with polyA-enrichment and NovaSeq X Plus Series Sequencing (paired end reads, ~70−100 M reads per sample) were performed by Novogene. Reads were aligned to the Hg19 human genome using Rsubread[60]. Feature count was performed using featureCounts[61], and differential expression and PCA analysis using DESeq2[62]. Differential gene expression threshold was set to adjusted p-value < 0.05. KEGG 2021 pathway analyzes and GO ontology analyzes were performed using Enrichr[63].

## Single-molecule fluorescence microscopy

Live cell protein labeling for single-molecule microscopy was done by labeling with a combination of 2 μM SNAP SiR-647 (New England Biolabs) and 1 μM Halo R110 (Promega) in complete culture medium for 20 min at 37 °C. Cells were then washed five times with complete culture medium, allowing 5 min incubation between washes.

Single-molecule microscopy experiments were performed using total internal reflection fluorescence (TIRF) illumination on a custom system (assembled by CAIRN Research) based on an Eclipse Ti2 microscope (Nikon, Japan) equipped with a 100x oil-immersion objective (SR HP APO TIRF NA 1.49, Nikon), 405, 488, 561, and 637 nm diode lasers (Coherent, Obis), an iLas2 TIRF illuminator (Gataca Systems), quadruple band excitation and dichroic filters, a quadruple beam splitter, 1.5x tube lens, four EMCCD cameras (iXon Ultra 897, Andor), hardware focus stabilization, and a temperature-controlled enclosure. The sample and objective were maintained at 37 °C throughout the experiments. Coverslips were mounted in a microscopy chamber filled with Hank's balanced salt solution (HBSS) supplemented with 10 mM HEPES, pH 7.5. A reduced oxygen environment (2–4% $O_2$) was provided in the imaging chamber to decrease photobleaching without increasing cytotoxicity using a mixture of nitrogen and air and a home-built gas mixing and humidifying system, similar as previously described[64]. Multi-color single-molecule image sequences were acquired (400 frames in length) simultaneously at a rate of one image every 33 ms.

Automated single-particle detection and tracking were performed with the u-track software[65] and the obtained trajectories were further analyzed using custom algorithms in MATLAB environment as previously described[42,66]. Image sequences from different channels were registered against each other, based on reference points taken with multi-color fluorescent beads (100 nm, TetraSpeck)[42]. The inter-channel localization precision after coordinate registration was ~20 nm. The time-averaged mean squared displacement (TAMSD) of individual trajectories as well as stepwise photobleaching were computed as previously described[42,59]. To obtain the diffusion coefficient (D), only trajectories lasting at least 100 frames were analyzed. For the stepwise photobleaching analysis, a step fitting of the intensity profile was calculated for each particle, setting the maximum step number as 15. Only co-confined molecules were analyzed. A spatial confinement analysis was used to identify trajectory segments characterized by confinement[67]. The frequency and duration of $FZD_5$–LRP6 interactions were estimated using previously described methods based on deconvolution of the distribution of single-molecule colocalization times with the one expected for random colocalizations[42]. Trajectory segments were first linked to obtain continuous trajectories that are no longer interrupted by merging and splitting events. Then, for each particle in the $FZD_5$ channel at frame f, all particles in the LRP6 channel falling within a defined search radius (150 nm) were identified as colocalizing. If a colocalization was also present at frame f + 1, the colocalization was extended. The process was iterated until the last frame of the image sequence. These data were used to build a matrix containing information for each colocalization (involved particles as well as the start and end frames). The observed colocalization time corresponds to the duration of true interactions plus the duration of random colocalizations. Thus, the distribution of the observed colocalization times can be seen as a convolution of the distribution of true interaction times and random colocalization times. The distribution for random colocalizations was estimated in cells co-transfected with Halo-LRP6 and SNAP-$\beta_2$AR, a membrane receptor with similar diffusion characteristics to that of $FZD_5$. To obtain the true colocalization time, deconvolution with the Lucy–Richardson algorithm was performed. The separation of co-diffusion versus co-confinement was estimated as previously described, based on whether the interacting partners ($FZD_5$ and LRP6 molecules) were diffusing on the plasma membrane or were confined during their interaction time[66].

## Reporting summary

Further information on research design is available in the Nature Portfolio Reporting Summary linked to this article.

## Data availability

The RNA sequencing data set is available at Genome Sequence Archive with the accession code HRA008938. Source data are provided with this paper.

## Code availability

All code used in this manuscript (R packages for the analysis of RNA sequencing data, Matlab code for analysis of single-particle tracking movies) has been published in the cited references.

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

## Acknowledgements

We thank Dr. Melissa Gammons (Cambridge Institute for Medical Research) for providing HEK293 ΔDVL1-3 cells, and Dr. Benoit Van-hollebeke (Université libre de Bruxelles) for providing HEK293 ΔFZD$_{1-10}$ and HEK293 ΔLRP5/6 cells. We thank S. Angers (University of Toronto) and V. Bryja (Brno University) for providing plasmids encoding MESD and DVL/RNF43 constructs, respectively. The work was supported by grants from Karolinska Institutet, the Swedish Research Council (GS: 2019-01190; 2024-02515), the Swedish Cancer Society (GS: 20 1102 PjF, 23 2825 Pj), the Novo Nordisk Foundation (GS: NNF22OC0078104), and the German Research Foundation (DFG; LG: 504098926; JHV: 520506488). This project has received funding from the Innovative Medicines Initiative 2 Joint Undertaking (JU) under grant agreement No 875510. The JU receives support from the European Union's Horizon 2020 research and innovation program and EFPIA and Ontario Institute for Cancer Research, Royal Institution for the Advancement of Learning McGill University, Kungliga Tekniska Högskolan, Diamond Light Source Limited. This communication reflects the views of the authors and the JU is not liable for any use that may be made of the information contained herein. DC was supported by a Wellcome Trust Senior Research Fellowship (212313/Z/18/Z). JTL acknowledges the Swedish Research Council and the Strategic Research Program in Diabetes for funding.

## Author contributions

Conceptualization: J.V., G.S.; methodology: all authors; data curation, formal analysis, and visualization: J.V., Z.K., E.S.; funding acquisition, project administration, and resources: G.S., D.C., J.T.L.; investigation: J.V., Z.K., Y.Y., L.G., E.S.; writing—original draft: J.V., Z.K., E.S., G.S.; writing – review & editing: J.V. and G.S. with contributions from all authors.

## Funding

## Competing interests

The authors declare no competing interests.
