## [Transparent Peer Review file · Nature Communications]

WNT-induced association of Frizzled and LRP6 is not sufficient for the initiation of WNT/ β -catenin signaling

Corresponding Author: Professor Gunnar Schulte

Version 0:

Reviewer comments:

Reviewer #1

(Remarks to the Author)

In this manuscript, Hendrik Voss et al. describe the activation process of WNT/ β -catenin signaling by employing BRET to monitor the interaction between FZD and LRP6. They propose a two-step model in which the initial ligand-induced association of FZD and LRP6 is followed by LRP6 phosphorylation and the clustering of receptors into higher-order complexes, which are essential for the effective activation of WNT/ β -catenin signaling. However, the paper contains significant issues regarding the experimental data, its presentation, and interpretation, which hinder its suitability for publication.

1. Traditional Wnt ligands, such as WNT-3A and WNT-16B, undergo lipid modifications essential for signaling and membrane anchoring. However, in this study, recombinant purified WNT-3A and WNT-16B were used as ligands, resulting in a limited BRET observation window of 0 to 0.02, which raises concerns about the reliability of the data.
2. Figure 3D shows that the WNT-16B-induced FZD-LRP6 association, as measured by BRET, is very weak, with values below 0.01. Similarly, the p-LRP6 band in Figure 3E is also faint, indicating low levels of LRP6 phosphorylation; however, it remains detectable, confirming that phosphorylation does occur.
3. The current data do not support the proposed statement that "WNT-3A and WNT-16B induce FZD5-LRP6 protein complexes with distinct architectures."
4. Do your data support the previously reported observation that 'WNT induces FZD clustering, followed by association with LRP5/6'?
5. Page 16. In the sentence, 'This observation was independent of DVL polymerization and may speculatively be explained by DVL acting as a steric hindrance, impeding optimal Nluc-Venus energy transfer, or serving as a scaffold for the FZD5 C-terminus,' does DVL indeed act as a steric hindrance that impedes energy transfer?

Reviewer #2

(Remarks to the Author)

This study aims to understand early events in activation of Wnt signaling, especially focusing on interactions between Fzd5 and LRP6. Using newly established BRET assays, the authors monitored associations of Fzd-LRP5/6 upon Wnt stimuli. In their intensive BRET assays, they effectively used several functionally-identified mutants of Fzd, LRP5/6 and Dvl2 with proper combinations of gene-knockout cells and clarified some elemental processes of signal reception upon Wnt3a stimulation. In the following sections, differential signaling between Wnt3a and Wnt16b are described. Their single molecule imaging-based analyses are carefully designed to support their claim of difference in higher order complex formation upon Wnt3a and Wnt16b stimuli. Although which molecular properties of these ligands are decisive for the differential Fzd5-LRP6 complex formation and the following phosphorylation of LRP6, the presented work provides several interesting data and potential tools to understand the entrance of Wnt signaling.

Overall, this report reflects significant efforts, and is highly original and would be of interest to the field. However, there are several issues that the authors are encouraged to address before publication is recommended.

Major issues

1. Their BRET assays provides much information about temporal response of Fzd and LRP upon stimulation with Wnt ligands. However, signaling capability of their probes (LRP6-Venus and Fzd5-Nluc) were not directly examined. Using these constructs to rescue Wnt/ β -catenin signaling in Δ LRP5/6 and Δ Fzd1-10 cells, respectively, upon Wnt3a stimulation is recommended to clarify these probes' function. In addition, the BRET readout was only provided in Δ BRET values. Is it possible to provide some micrographs of BRET activation with proper microscopy? If the BRET signal can be detected after weak fixation, comparisons between BRET and immunostaining of phospho-LRP6 or -Dvl2 would provide informative data of connection between the BRET signal and direct downstream events (related to 2).
2. Based on immunoblotting, Figures 3E-G provide interesting data of different phosphorylation of LRP6 and Dvl2 upon stimulations with several different Wnt ligands. However, sensitivity of immunoblotting could be lower than that of immunostaining, and immunostaining with these phospho-specific antibodies could provide more information such as clustering and/or colocalization of these components.
3. The authors claim that Wnt16b can stimulate Fzd-LRP6 association without stabilizing β -catenin. The association of Fzd and LRP6 was firmly supported by their data, however, stabilization of β -catenin was not directly examined. The TOPFlash, which is based on multiplex TCF/LEF binding sites, is generally used as a good reporter of Wnt/ β -catenin signaling, but their activation could be context-dependent. Thus, direct evidence of β -catenin stabilization should be provided.
4. Figure 4 provides very interesting datasets. An interesting point is that some of genes upregulated by Wnt3a (these are usually regarded as targets of Wnt/ β -catenin signaling) are also upregulated by Wnt16b, even though Wnt16b does not seem to activate Wnt/ β -catenin signaling. However, detailed annotations are lacking. For example, a Venn diagram of those target genes with gene ontology analyses would be helpful for the Wnt community as well as broader readers. Analyses of TCF/LEF binding sites in the vicinity of upregulated genes and/or ChIP-seq of β -catenin would clarify their findings and claim (related to 3). Given that many genes seemed to be commonly upregulated and downregulated by both Wnt3a and Wnt16b, the difference between the genes specifically responded by either of the Wnt3a or Wnt16b should be analyzed. With proper annotations, the provided data might dissect so-called "Wnt/ β -catenin signaling", for example, into target genes commonly activated by Wnt3a and Wnt16b and truly β -catenin-dependent ones. In addition, β -catenin independent functions of Wnt/LRP6 signaling have been described (PMID: 24837680, 27568239). Such kinds of signaling could be discussed in this work.

Minor issues

In schematic illustrations, the shape of Nluc and Venus are identical. I felt that it is a little bit confusing because they are not structurally related.

The current title may have some impact on the Wnt community, however, it reflects only a part of the content.

Reviewer #3

(Remarks to the Author)

In this study, the authors described the association of Lrp6 and Fzd in the presence of Wnt16b, which did not lead to Lrp6 phosphorylation or activation of the Topflash activity assay. Previous studies have demonstrated that Wnt16b can signal through both β -catenin-dependent pathways (refs 46, 47) and β -catenin-independent pathways (Zhao et al., Stem Cell Reports, 2022, DOI: 10.1016/j.stemcr.2022.03.001). Therefore, binding of Wnt16b to Lrp6 is not surprising. The authors speculate that the observed effects might be tissue-specific or context-dependent, potentially influenced by the presence of co-factors or specific Frizzled paralogs. Is it known which frizzled were tested in the previous studies cited? Testing different Frizzled receptors in HEK293 cells in this study would provide critical insight into the mechanisms involved and observation made in this study.

2. The authors showed that Wnt10b, which typically signals through the β -catenin pathway, did not activate the Topflash assay in this study. They speculate that this outcome might depend on the presence of other Frizzled paralogs. If this is the case, experiments should be performed using the appropriate Frizzled paralogs to test this hypothesis.

3. Wnt proteins are notoriously challenging to purify and tend to aggregate in solution. The authors should demonstrate the purity and activity of the Wnt ligands used in this study after reconstitution. The use of afamin, which has been shown to stabilize Wnt ligands, could further validate the results.

4. In contrast to Wnt10b, Wnt5a, which typically activates β -catenin-independent pathways, demonstrated some BRET activity similar to Wnt16b. Including Wnt5a as a control and comparison in the single-molecule fluorescence study would strengthen the findings and provide valuable context.

5. The bulk RNA-seq data from HEK293 cells stimulated with Wnt3a did not show upregulation of canonical Wnt target genes such as Axin2 and Notum. The authors should compare their data with previously published datasets and provide an explanation for the observed discrepancy.

6. For WNT16b-stimulated cells, the authors could determine which pathways are upregulated, e.g., based on GSEA analysis and comparing the results with previous work, such as that by Zhao et al. This would provide additional insights into the mode of action of WNT16b in this study.

7. The authors claimed that the orientation of the FZD5 and LRP6 C-termini is allosterically modulated by DVL binding to FZD, resulting in a reduced WNT-induced Δ BRET_{max}. However, Feng Cong's work (Molecular Cell, 2015) demonstrated that DVL knockout significantly increases the surface expression of Fzd and LRP6 and mediates the degradation of FZD

through ZNRF3. Could this mechanism explain the observations made here? This should be experimentally tested.

8. Related to 7, Frizzled receptors are known to be targeted for degradation by ZNRF3, particularly in the presence of Wnt ligands. Some experiments in this study should be performed in the presence of R-spondin.

9. In Figure 3, the Topflash assays were performed after 24 hours of Wnt stimulation, whereas the BRET assays were performed within 60 minutes. The authors should include staining for phosphorylated β -catenin and total β -catenin within the same timeframe.

Version 1:

Reviewer comments:

Reviewer #1

(Remarks to the Author)

The authors have adequately addressed my comments and critiques, making the work suitable for publication.

Reviewer #2

(Remarks to the Author)

In the revised manuscript, the authors properly addressed my comments, including functional validation of engineered receptor constructs and protein levels of beta-catenin (total and phosphorylated) in response to Wnt3a and Wnt16b. Additional data using a phosphorylation-deficient LRP6 (LRP6-5A) also provides insights into assembly of the signalosomes. Overall, this study would provide an important data set to understand differences among Wnt ligands and their signaling activity. Thus, this study may now be accepted for publication in Nature Communications.

Reviewer #3

(Remarks to the Author)

The reviewer appreciates the additional data the authors have provided to clarify the technical aspects of this study. However there remain concerns about whether the overarching conclusions are fully supported by the data presented in this study.

The authors present the central finding that WNT16b induces the association of FZD5 and LRP6 but does not activate β -catenin–dependent signaling. They propose that WNT16b challenges the idea that β -catenin activation necessarily follows from ligand-induced FZD–LRP6 interactions, an assumption commonly supported by the use of WNT surrogates, which act as extracellular crosslinkers to potentially trigger WNT/ β -catenin signaling.

Conceptually, however, this finding is not entirely novel. Previous reports have demonstrated that non-canonical WNTs (e.g., WNT5a) can engage LRP6 without activating β -catenin mediated signaling (Sato et al., 2009, EMBO J.; Bryja et al., 2009, Mol. Biol. Cell).

The authors provide extensive evidence showing that WNT16b does not induce LRP6 phosphorylation or β -catenin–driven transcription, consistent with earlier findings indicating that WNT16b engages non-canonical pathways (Teh et al., 2007, J. Cell Sci.). The BRET assay showed interaction for WNT16b–LRP6 but not WNT5a–LRP6. It is not clear why the two non-canonical ligands behave differently. It is important to note that HEK293 cells have high endogenous LRP6/FZD and are typically used for β -catenin mediated TOPFlash activity. Therefore, detecting a WNT16b–LRP6 interaction is not completely unexpected and does not necessarily reflect a broader physiological context. Furthermore, the authors also indicated the overexpression of FZD/LRP lead to the high levels of surface expression, and lead to the saturation of ubiquitination by RNF43. The nature of LRP6 and WNT16b interaction reported here requires further clarifications. Can WNT16b signal in the absence of LRP6? Could WNT16b interaction with LRP6 be stabilized by other cofactors, not present in HEK293 cells? In previous studies involving Wnt16b, other cell types have been used.

A major limitation of the overarching conclusion of this study ‘WNT-induced association of Frizzled and LRP6 is not sufficient for the initiation of WNT/ β -catenin signaling’ is primarily based on one Wnt ligand, i.e., WNT16b. In many cases, e.g., WNT3a, ligand-induced FZD–LRP6 interaction does lead to clustering and initiate robust β -catenin signaling. Similarly, WNT surrogates can crosslink these receptors and drive canonical WNT activity, even when acting through FZD isoforms thought to be non-canonical. This study did not investigate whether Wnt surrogate initiates higher order clustering, a critical gap in the analysis.

The observation that “initial ligand-induced FZD–LRP6 association must be followed by receptor clustering into higher-order complexes and subsequent phosphorylation of LRP6 for efficient activation” has been long recognized in the field (e.g., Bilic, Science, 2007). The author observed clustering with single particle tracking for WNT3a but not WNT16b, which also fail to activate β -catenin signaling. The study did not directly show that monomeric WNT3A–Lrp6–Fzd interaction is insufficient for β -catenin signaling. Rather the conclusions are largely based on the observation that Wnt16b that interact with LRP6 but did not form higher order clusters or activate WNT signaling.

Other comments:

The RNA-seq data for both Wnt3a and Wnt16b are somewhat inconclusive. The authors claimed that HEK293 cells may not be optimal for studying WNT signaling but this is puzzling. RNA-seq data and other experiments such as single particle tracking were not performed in the presence of R-spondin, even though the TOP-flash assay showed marked increase in activity in the presence of WNT3a/R-spondin. Likewise, HEK293 may not be optimal for studying b-catenin independent signaling.

REBUTTAL LETTER

We are very thankful for the overall positive comments and constructive criticism. We think that your input indeed improved our manuscript. Below you find a detailed point-by-point rebuttal, where our author comments are listed in green.

The main changes include:

1. New experiments with R-spondin 1 co-stimulation and RNF43 wt and mutant co-expression to elucidate the contribution of FZD ubiquitination in our experimental system.
2. Detection of β -catenin protein levels and phosphorylation status (immunoblotting) in response to WNT-3A and WNT-16B.
3. New data for the single particle tracking employing FZD5 in combination with a phosphorylation-site-deficient LRP6 (LRP6-5A) excluding the need for LRP6 phosphorylation in the WNT-3A-induced formation of higher-order clustering.
4. Thorough functional validation of the engineered receptor constructs.

Reviewer #1 (Remarks to the Author)

In this manuscript, Hendrik Voss et al. describe the activation process of WNT/ β -catenin signaling by employing BRET to monitor the interaction between FZD and LRP6. They propose a two-step model in which the initial ligand-induced association of FZD and LRP6 is followed by LRP6 phosphorylation and the clustering of receptors into higher-order complexes, which are essential for the effective activation of WNT/ β -catenin signaling. However, the paper contains significant issues regarding the experimental data, its presentation, and interpretation, which hinder its suitability for publication.

Thanks for the efforts put into the review of our work. The reviewer raises “significant issues regarding experimental data, its presentation, and its interpretation”. We have addressed the reviewer’s concerns, but we do not feel that the criticism below provides enough detail to enable a constructive revision process.

1. While the statement that WNTs require lipidation in order to function is correct, the claim that the employed WNTs are not lipidated is not justified by any evidence. The activity of recombinant WNTs is thoroughly demonstrated in this study and in various other studies employing recombinant WNTs.
2. No criticism of data presentation is provided by this reviewer despite its mention in the introduction as a reason to hinder publication of our work.
3. The reviewer criticizes our data interpretation. However, except for mentioning the assay window of our BRET assays and asking for extension of our current

interpretation, no example of misinterpretation of our results is provided. In short, we do not know which part of our data we might have misinterpreted.

Below, we are addressing the reviewer's points in detail but are left with uncertainty about the substance of the criticism.

1. Traditional Wnt ligands, such as WNT-3A and WNT-16B, undergo lipid modifications essential for signaling and membrane anchoring. However, in this study, recombinant purified WNT-3A and WNT-16B were used as ligands, resulting in a limited BRET observation window of 0 to 0.02, which raises concerns about the reliability of the data.

We believe that this point is the reviewer's main point of concern.

We have used purified WNTs for 20 years adding new levels of quantitative assessments of ligand-receptor interaction and dynamics in diverse cell systems¹⁻⁵. We acknowledge that endogenously secreted WNTs act in a way that we do not fully grasp yet (e.g., WNT delivery by carrier proteins). However, we know that using recombinant purified WNTs allows us to study WNT signaling with a greater level of control than traditional experiments performed with conditioned media, whose exact composition (in particular of the secreted WNTs) is always unknown and can vary from batch to batch. There are 19 mammalian WNTs, which can trigger differential signaling pathways, as most prominently shown for WNT-3A and WNT-5A. Here, we set out to dissect differences between WNT-3A- and WNT-16B-induced signaling employing state-of-the-art methodology and innovative combination of knock out cells and experimental paradigms. It is not justified to blame these intrinsic differences of the WNTs on the surmised lack of a protein lipidation, especially when there is no evidence for the claim and when the experiments are perfectly controlled showing that – in this case WNT-16B – elicits a FZD-mediated DVL shift and FZD-LRP6 complex formation, effects that can only be explained by a lipid-mediated WNT-FZD interaction

The WNTs that were used in our work are produced from mammalian cells (Chinese hamster ovary) and are lipidated. In fact, we have shown for WNT-3A and WNT-5A (not WNT-16B) already in 2005 that enzymatic removal of the lipidation blunts their biological activity¹. Furthermore, the WNTs employed here elicit biologically relevant changes such as LRP6 phosphorylation, DVL shift and TOPFlash underlining their functionality. In fact, recombinant WNT-3A elicited a TOPFlash signal resembling that of highly efficacious WNT surrogates. Furthermore, both recombinant WNTs led to changes in the cellular transcriptome, proving their activity in an unbiased fashion. For WNT-16B, we denatured the WNT with heat inactivation, and consequently it became inactive in our BRET assay, further emphasizing functionality of the WNTs and the specificity of our BRET assay (Fig. S7).

We could list a long list of assay formats that were used to assess recombinant WNT activity outside of this publication ranging from immunoblotting, activation of heterotrimeric G proteins, biosensors and transcriptional reporter assays. For a very early example of an investigation of recombinant WNT effects on a microglia cell line see ³. Here, most of the WNTs would not elicit β -catenin-dependent WNT signaling but activate other pathways. These findings are also not explainable by a lack of WNT lipidation. Specifically, for WNT-16B we observed activation of a unimolecular FZD-DEP BRET biosensor to the same degree as WNT-3A in multiple human FZD paralogs ⁶, and others employed recombinant WNT-16B for *in vivo* wound healing assays ⁷ and biochemical readouts as well as reporter gene assays ⁸.

Regarding the Δ BRET amplitude, we would like to point out that this amplitude is first and foremost depending on the biosensors. Thus, the absolute amplitude of the assay is not directly related to the relevance of a biological effect. In this case, we measured BRET between two proteins of the same compartment (plasma membrane), and these setups typically have a narrower dynamic range, compared for example to recruitment assays where a cytosolic protein is recruited to the plasma membrane. Moreover, FZDs are notoriously challenging to investigate. In our experience, the amplitude of the BRET responses measured here is substantial and it is normal that the responses present with different ranges depending on the different experimental paradigms. The positive controls for FZD-LRP interaction are WNT-3A and the WNT surrogate, which routinely elicit a Δ BRET of 0.01-0.03. This is the biologically relevant assay window of the experiments performed. Provided our longstanding experience in the use of genetically encoded biosensors and BRET assays, and the fact that all values from Δ BRET traces in question are clearly different from 0 (i.e., no window), we do not understand how the reliability of the data can be questioned here.

2. Figure 3D shows that the WNT-16B-induced FZD-LRP6 association, as measured by BRET, is very weak, with values below 0.01. Similarly, the p-LRP6 band in Figure 3E is also faint, indicating low levels of LRP6 phosphorylation; however, it remains detectable, confirming that phosphorylation does occur.

The response on FZD-LRP BRET elicited by WNT-16B in Fig. 3D roughly represents that from WNT-3A, which can be seen as a positive control, in this series of experiments (Fig. 3A). The observed Δ BRET increase is reliable and reproducible (N=3).

We do not understand the question of this reviewer regarding the LRP6 phosphorylation in response to WNT-16B. We do not argue that WNT-16B reduces LRP6 phosphorylation below basal, and the reviewer will appreciate that the P-LRP6 band in the vehicle control is not significantly lower (in any case, it is rather larger) than that of WNT-16B, therefore we argue that WNT-16B-dependent LRP6 phosphorylation does either not take place

under these experimental conditions, or is below the limit of detection for immunoblotting.

3. The current data do not support the proposed statement that “WNT-3A and WNT-16B induce FZD5-LRP6 protein complexes with distinct architectures.”

Our work deals with the experimental differentiation between FZD-LRP interaction and the formation of higher order clusters. The single particle tracking experiments are stunningly defining the ability of WNT-3A to induce higher order complexes, whereas WNT-16B does not lead to the formation of these clusters despite the fact that both WNTs are able to promote FZD-LRP interaction (see BRET assay results in Fig. 3 and single-molecule tracking results in Fig. 4). We think that a higher order of receptor complexes can be described as “receptor complex architecture”. Our data is, however, not providing information on the conformation of a single FZD-WNT-LRP unit, though. Therefore, we see that the word “architecture” can be unclear, and we revised the subheading in question (“WNT-3A and WNT-16B differ in their capacity to induce higher-order FZD₅-LRP6 clusters”).

4. Do your data support the previously reported observation that 'WNT induces FZD clustering, followed by association with LRP5/6?

We are unaware of the data referred to by the reviewer. We are however aware of the data on FZD dimerization, where both ligand-independent and ligand-induced FZD dimerization and de-dimerization were reported by us and others^{2,9,10}, but which we did, notably, not observe for FZD₅¹¹. Nevertheless, we see the FZD-FZD dimerization as a separate phenomenon from FZD-LRP association and clustering.

5. Page 16. In the sentence, 'This observation was independent of DVL polymerization and may speculatively be explained by DVL acting as a steric hindrance, impeding optimal Nluc-Venus energy transfer, or serving as a scaffold for the FZD5 C-terminus,' does DVL indeed act as a steric hindrance that impedes energy transfer?

In the statement that the reviewer cites, we specifically use careful formulations as we do not know for sure, but our data clearly shows that DVL binding to FZD results in less optimal resonance energy transfer between the respective tags at the FZD₅ and LRP6 C-termini. Therefore, we think it is justified to say that DVL acts as a steric hindrance. While we see our DVL-related data as highly relevant and significant in the context of higher order receptor complex formation, we have reached the technological limitations to dissect the observed phenomena more. Therefore, we are forced to present a speculative interpretation, and future work will address this question in more detail. A further investigation is at this point out of the scope of this work.

As this sentence may be unclear, we rephrased the sentence for the sake of clarity (“This observation was independent of DVL polymerization. We speculate that DVL binding to FZD may induce a conformation of the FZD C-terminus which is not optimal for Nluc-Venus energy transfer”).

Reviewer #2 (Remarks to the Author)

This study aims to understand early events in activation of Wnt signaling, especially focusing on interactions between Fzd5 and LRP6. Using newly established BRET assays, the authors monitored associations of Fzd-LRP5/6 upon Wnt stimuli. In their intensive BRET assays, they effectively used several functionally-identified mutants of Fzd, LRP5/6 and Dvl2 with proper combinations of gene-knockout cells and clarified some elemental processes of signal reception upon Wnt3a stimulation. In the following sections, differential signaling between Wnt3a and Wnt16b are described. Their single molecule imaging-based analyses are carefully designed to support their claim of difference in higher order complex formation upon Wnt3a and Wnt16b stimuli. Although which molecular properties of these ligands are decisive for the differential Fzd5-LRP6 complex formation and the following phosphorylation of LRP6, the presented work provides several interesting data and potential tools to understand the entrance of Wnt signaling.

Overall, this report reflect significant efforts, and is highly original and would be of interest to the field. However, there are several issues that the authors are encouraged to address before publication is recommended.

Thanks a lot for the positive comments regarding the significance of our work and for the constructive criticism, which we are addressing in detail in the sections below.

Major issues

1. Their BRET assays provides much information about temporal response of Fzd and LRP upon stimulation with Wnt ligands. However, signaling capability of their probes (LRP6-Venus and Fzd5-Nluc) were not directly examined. Using these constructs to rescue Wnt/ β -catenin signaling in Δ LRP5/6 and Δ Fzd1-10 cells, respectively, upon Wnt3a stimulation is recommended to clarify these probes' function. In addition, the BRET readout was only provided in Δ BRET values. Is it possible to provide some micrographs of BRET activation with proper microscopy? If the BRET signal can be detected after weak fixation, comparisons between BRET and immunostaining of phospho-LRP6 or -Dvl2 would provide informative data of connection between the BRET signal and direct downstream events (related to 2).

Thanks for this relevant question. We have added new data in the supplementary information (see Supp. Fig. 1D+E) to pinpoint the functionality of all receptor constructs

that were used as BRET sensors and during single particle tracking experiments, including LRP5/6-Venus, Halo-LRP6, FZD₅-Nluc and SNAP-FZD5, using the TOPFlash assay. However, the visualization of BRET by microscopy is not reliably possible. BRET microscopy is technically extremely challenging, and would require both specific instrumentation and also extremely high Δ BRET amplitudes – far higher than those yielded from our sensors. Unfortunately, while highly interesting, this is not possible with our sensors and available equipment.

2. Based on immunoblotting, Figures 3E-G provide interesting data of different phosphorylation of LRP6 and Dvl2 upon stimulations with several different Wnt ligands. However, sensitivity of immunoblotting could be lower than that of immunostaining, and immunostaining with these phospho-specific antibodies could provide more information such as clustering and/or colocalization of these components.

In our experience, the immunoblotting analysis is generally more sensitive and quantifiable to assess WNT-induced changes in any of the hallmarks of WNT/ β -catenin signaling, with the exception, maybe, of β -catenin nuclear translocation. Immunoblotting readily allows analysis of WNT-induced biochemical changes addressing endogenously expressed proteins. When using immunocytochemistry one always has to struggle with the risk for unspecific signal, which can – at least in part – be reduced in immunoblotting given the size separation of the bands. For examples where we have used immunocytochemistry for assessing changes in WNT signalling see the following papers ^{4,12}. Furthermore, detection of endogenous DVL by immunocytochemistry has not been possible reliably, and especially the dynamic WNT-induced recruitment of endogenous DVL has evaded detection so far. On the same note, we have previously tried to utilize the commercially available anti-P-LRP6 antibody, which we employed in Western blotting, in immunocytochemistry without success. Nevertheless, we agree with the reviewer's comment. For the above-mentioned technical reasons, the suggested experiments could not be done.

3. The authors claim that Wnt16b can stimulate Fzd-LRP6 association without stabilizing β -catenin. The association of Fzd and LRP6 was firmly supported by their data, however, stabilization of β -catenin was not directly examined. The TOPFlash, which is based on multiplex TCF/LEF binding sites, is generally used as a good reporter of Wnt/ β -catenin signaling, but their activation could be context-dependent. Thus, direct evidence of β -catenin stabilization should be provided.

In order to address this comment, we have compared WNT-3A and WNT-16B with regard to their effects on total β -catenin levels and β -catenin phosphorylation in immunoblotting experiments (Supp. Fig. 6B-E). The new data on β -catenin are well in line with our TOPFlash data and underscore that WNT-16B does, in contrast to WNT-3A,

not elevate total β -catenin levels and does not decrease the amount of phosphorylated (i.e. marked-for-degradation) β -catenin.

4. Figure 4 provides very interesting datasets. An interesting point is that some of genes upregulated by Wnt3a (these are usually regarded as targets of Wnt/ β -catenin signaling) are also upregulated by Wnt16b, even though Wnt16b does not seem to activate Wnt/ β -catenin signaling. However, detailed annotations are lacking. For example, a Venn diagram of those target genes with gene ontology analyses would be helpful for the Wnt community as well as broader readers. Analyses of TCF/LEF binding sites in the vicinity of upregulated genes and/or ChIP-seq of β -catenin would clarify their findings and claim (related to 3). Given that many genes seemed to be commonly upregulated and downregulated by both Wnt3a and Wnt16b, the difference between the genes specifically responded by either of the Wnt3a or Wnt16b should be analyzed. With proper annotations, the provided data might dissect so-called “Wnt/ β -catenin signaling”, for example, into target genes commonly activated by Wnt3a and Wnt16b and truly β -catenin-dependent ones. In addition, β -catenin independent functions of Wnt/LRP6 signaling have been described (PMID: 24837680, 27568239). Such kinds of signaling could be discussed in this work.

We are happy that the reviewer finds this data set interesting. However, it cannot be emphasized sufficiently that the intention of these experiments was to provide an unbiased readout for WNT-3A and WNT-16B activity. The intention was not to employ the rather non-physiological cell system of HEK293 cells to analyze and dissect WNT-induced signaling by transcriptomics. Nevertheless, we have followed up on the reviewer’s comments as detailed below.

We have analyzed TCF/LEF binding sites in the vicinity of differentially expressed genes, but due to the short and variable sequence of the TCF/LEF binding sequence, and a quite long range in which promoters can bind upstream to the transcriptional start (~2 kb), we identified such sites in roughly two thirds of all human genes, which was not enriched in the group of WNT-3A-induced genes. We have also added a statement on β -catenin-independent functions of WNT/LRP6 signaling to the discussion.

We have avoided the detailed annotation of the genes to argue mostly in an unbiased manner that both WNTs show effects on gene transcription that differ and that could explain the differences in the receptor dynamics elicited by WNT-3A and -16B. While we agree with reviewers 2 and 3 that the RNA-seq data would be of interest to the field, we are cautious in order to not overinterpret our data as HEK293 cells are not an optimal model system for physiological WNT readouts. We suspect that our cells have a high basal β -catenin-like signaling, suggested by a well-detectable P-LRP6 band in vehicle-

treated cells at basal, and the fact that WNT-3A-treated samples clustered quite closely with vehicle-treated samples.

Additionally, as Reviewer 3 also pointed out, no established β -catenin target genes are upregulated by WNT-3A treatment in our RNA-seq results, despite strong activation of WNT/ β -catenin signaling as assessed by TOPFlash and Western blotting. A similar RNA-seq approach in corneal stem cells⁷ identified chemokine receptor signaling as an effector pathway of WNT-16B, which we could not verify in HEK293 cells either.

We have, however, added descriptive statistics of our analyses to the Figure panel, but also added a statement of caution not to interpret RNA-seq findings from HEK293 cells as a proper reflection of physiological WNT target genes. We deem our RNA-seq experiments appropriate for our intent to show that WNT-3A and WNT-16B are both active and feed into different pathways, but strongly advocate against interpreting the differentially expressed genes in HEK293 cells as a blueprint for human physiology. To underscore this, we moved the respective figures to supplementary information.

Minor issues

In schematic illustrations, the shape of Nluc and Venus are identical. I felt that it is a little bit confusing because they are not structurally related.

Thanks for this constructive observation. We agree and have changed the relevant schemes to match the structure of the Nanoluciferase.

The current title may have some impact on the Wnt community, however, it reflects only a part of the content.

We have to admit that formulating a title that is catchy and covers the most important aspects of the paper is difficult. We have run another “creative” round of title discussions and ended up in not changing the current title that conveys the essence of our findings.

Reviewer #3 (Remarks to the Author):

We thank the reviewer for the thorough analysis and constructive criticism.

1. In this study, the authors described the association of Lrp6 and Fzd in the presence of Wnt16b, which did not lead to Lrp6 phosphorylation or activation of the Topflash activity assay. Previous studies have demonstrated that Wnt16b can signal through both β -catenin-dependent pathways (refs 46, 47) and β -catenin-independent pathways (Zhao et al., Stem Cell Reports, 2022, DOI: 10.1016/j.stemcr.2022.03.001). Therefore, binding of Wnt16b to Lrp6 is not surprising. The authors speculate that the observed effects might be tissue-specific

or context-dependent, potentially influenced by the presence of co-factors or specific Frizzled paralogs. Is it known which frizzled were tested in the previous studies cited? Testing different Frizzled receptors in HEK293 cells in this study would provide critical insight into the mechanisms involved and observation made in this study.

To our knowledge, previous studies did not test the effect of WNT-16B on specific FZD paralogs, but rather in tissue or cellular contexts. In the initial submission, we have tested untransfected HEK293 cells and individually assayed FZD_{4, 5, and 7}. In the revised version we have tested all remaining FZDs (1,2,8,10) that are described to activate WNT/ β -catenin signaling, and found a similar fingerprint: All these receptors activate the TOPFlash assay when stimulated with WNT-3A, but none of them is stimulated by WNT-16B. We therefore do not expect FZD paralog-selective activation of WNT/ β -catenin signaling via WNT-16B as a likely mechanism. We have adapted our statement in the discussion accordingly.

2. The authors showed that Wnt10b, which typically signals through the β -catenin pathway, did not activate the Topflash assay in this study. They speculate that this outcome might depend on the presence of other Frizzled paralogs. If this is the case, experiments should be performed using the appropriate Frizzled paralogs to test this hypothesis.

We have investigated whether WNT-10B acts via FZD_{1/2/8/10} instead, and detected no TOPFlash signal with any of these receptors. Considering that the resulting Δ BRET values obtained with WNT-10B were notably lower than with WNT-3A and WNT-16B, and the fact that WNT-10B did not cause an upward shift of DVL2 (which we interpret as evidence for WNT-FZD interaction), we conclude that WNT-10B does not employ the mechanisms covered by the assay methodology. In previous work, we have, however, used assays that reported on WNT-10B efficacy including FRAP measurements with FZD6-GFP and FZD-cpGFP conformational sensors^{5,13}.

3. Wnt proteins are notoriously challenging to purify and tend to aggregate in solution. The authors should demonstrate the purity and activity of the Wnt ligands used in this study after reconstitution. The use of afamin, which has been shown to stabilize Wnt ligands, could further validate the results.

We agree with the reviewer that purification of WNTs is challenging. We purchased purified WNTs from biotechne and have long-standing experience with purified

WNTs. The manufacturer sells WNTs with a minimum of 75% purity, and we use WNT preparations within the recommended shelf-life and strictly on ice to prevent damage to the samples. Particularly for WNT-16B we are convinced of its activity due to a strong response in BRET and DVL2 phosphoshift, and also in RNA-seq and single-molecule tracking. Furthermore, heat-inactivation of WNT-16B abolished its signal in BRET assays (Fig. S7). In summary we consider that we have thoroughly tested the activity of WNT preparations, in particular for WNT-3A and WNT-16B, which are crucial to this manuscript.

4. In contrast to Wnt10b, Wnt5a, which typically activates β -catenin-independent pathways, demonstrated some BRET activity similar to Wnt16b. Including Wnt5a as a control and comparison in the single-molecule fluorescence study would strengthen the findings and provide valuable context.

While we agree to the reviewer's comment, we have decided not to perform single-molecule tracking experiments with WNT-5A as we think that these experiments would divert attention from the direct comparison between WNT-16B and WNT-3A. While WNT-5A generally does not feed into the β -catenin pathway, the underlying mechanisms of how WNT-3A and WNT-5A differ in their mode of action remain still obscure. On one hand, WNT-5A would be a good control, on the other hand, the use of WNT-5A would open a completely different level of complexity. Our present results unambiguously show that WNT-16B promotes interaction between both proteins. This comment, however, inspired us to investigate the sequence of events in the initiation of WNT/ β -catenin signaling by utilizing the phosphorylation site LRP6-5A mutant in single-molecule tracking experiments to investigate whether LRP6 phosphorylation is a prerequisite for or a consequence of LRP6 phosphorylation (see Fig. 5).

5. The bulk RNA-seq data from HEK293 cells stimulated with Wnt3a did not show upregulation of canonical Wnt target genes such as Axin2 and Notum. The authors should compare their data with previously published datasets and provide an explanation for the observed discrepancy.

As the reviewer rightly pointed out, we did not observe any well-described WNT target genes, despite activation of WNT/ β -catenin signaling as assessed by TOPFlash and Western blotting (P-LRP6, P- β -catenin, total β -catenin). One explanation could be high basal WNT-3A like signal as suggested by clearly detectable P-LRP bands in vehicle-controlled samples (Fig. 3E), and rather proximate clustering of vehicle-treated and WNT-3A samples in the PCA plot of RNA-

seq results (Supp. Fig. 8). QPCR experiments previously performed in our group (attached below) also detected no transcriptional changes for *AXIN2* and *LGR5* in HEK293 cells, repetition of the same experiment in mouse embryonic fibroblasts, however, displayed clear induction of *Axin2* by WNT-3A (Kinsolving et al, Fig. 3D). Additionally, we could not detect activation of chemokine-chemokine receptor signaling by WNT-16B similar to Zhao et al.⁷, and feeding lists of differentially regulated genes obtained from RNA-seq into KEGG pathway analysis or gene ontology analysis yielded no significant results (see next point). We conclude that HEK293 cells are very likely poor model cells for WNT signaling on the transcriptional level: While they show all proximal hallmarks of WNT/ β -catenin signaling when stimulated, their transcriptional response does not match data obtained from more physiological systems. We have expanded on this issue in the discussion section and moved the RNA-seq data to the supplement. See also reviewer 2, point 4.

HEK293 cells show no significant WNT-3A induced regulation of the WNT target genes *AXIN2* and *LGR5*. Log₂-fold change of mRNA induction as assessed by qPCR. HEK293 cells were starved and treated with C59 for 24 hours before addition of 200 ng/ml WNT-3A (similar to our conditions in RNA-seq; here we used identical incubation times, but 300 ng/ml WNT-3A). Statistical analysis was performed by multiple t-tests (unpaired, two-tailed).

6. For WNT16b-stimulated cells, the authors could determine which pathways are upregulated, e.g., based on GSEA analysis and comparing the results with previous work, such as that by Zhao et al. This would provide additional insights into the mode of action of WNT16b in this study.

We have performed KEGG pathway analysis as in Zhao et al.⁷, and also performed gene ontology analyses. Curiously, we observed no significant ($\text{padj} < 0.05$)

regulation of any pathway in any analysis using either genes uniquely regulated by WNT-3A, uniquely regulated by WNT-16B, commonly regulated (either direction) or commonly up/downregulated (each direction tested separately).

7. The authors claimed that the orientation of the FZD5 and LRP6 C-termini is allosterically modulated by DVL binding to FZD, resulting in a reduced WNT-induced Δ BRET_{max}. However, Feng Cong's work (Molecular Cell, 2015) demonstrated that DVL knockout significantly increases the surface expression of Fzd and LRP6 and mediates the degradation of FZD through ZNRF3. Could this mechanism explain the observations made here? This should be experimentally tested.

In Supp. Fig. 4A, we have tested whether FZDs are upregulated in Δ DVL cells and found the surface expression of recombinantly expressed HA-FZD₅-Nluc unaffected by the DVL knockout and subsequent DVL2 rescue experiments using the same transfection conditions as in the BRET assays. We have added the corresponding data for LRP6 in Δ DVL cells and similarly observed no changes in receptor surface expression and conclude that differential receptor surface expression cannot be the mechanism behind our observations. We hypothesize that recombinant FZD/LRP expression under a strong promoter might oversaturate the endogenous FZD ubiquitination system. See also the response to the next point.

8. Related to 7, Frizzled receptors are known to be targeted for degradation by ZNRF3, particularly in the presence of Wnt ligands. Some experiments in this study should be performed in the presence of R-spondin.

We thank the reviewer for this suggestion and agree with their concern regarding an impact of receptor ubiquitination/internalization and degradation on our results.

In reporter gene assays, we have directly compared the effect of recombinant R-spondin 1 on TOPFlash elicited by WNT-3A and WNT-16B stimulation (Fig. 3I). Here, we observed a strong boost in the TOPFlash signal when RSPO1 was added together with WNT-3A into HEK293 cells, but not when it was added in combination with WNT-16B. We therefore conclude that a potential WNT-16B signal is not blunted by FZD internalization.

Following up, FZD₅, which was used as a prototypic FZD in WNT/ β -catenin signaling throughout this study, is mainly marked for degradation by RNF43 as shown in the work cited by the reviewer¹⁴ and additionally by the group of Madelon Maurice¹⁵. To probe whether RNF43-mediated ubiquitination affects our BRET assay, we additionally performed WNT-3A stimulation experiments monitoring BRET between FZD₅-Nluc and LRP6-Venus (i) after pre-incubation with RSPO1, and (ii) while co-

expressing wt RNF43 or WNT-hyperactivating mutants of RNF43. We observed that RSPO1 pre-incubation had no effect on the outcome of our BRET assay, while co-expression of wt (but not mutant) RNF43 severely blunted the BRET signal. This could not be rescued with RSPO1 pre-treatment. These results strengthen our hypothesis that FZD ubiquitination systems endogenous to HEK293 cells are insufficient to promote quantitative degradation of overexpressed FZD. Overexpression of RNF43, however, leads to a very quick attenuation of the BRET signal after WNT-3A treatment as suggested by the reviewer. The new results are included in Supp. Fig. 5. We have repeated the same set of experiments with WNT-16B as well (see below), yielding virtually identical results.

RNF43 co-expression blunts BRET assessment of FZD₅-LRP6 association independent of R-spondin 1 treatment. **A.** Kinetic Δ BRET traces and average Δ BRET values of HEK293 cells transfected with FZD₅-Nluc and LRP6-Venus. Cells were pre-incubated for 2 h with either vehicle or 100 ng/ml R-spondin 1, and subsequently stimulated with 1000 ng/ml WNT-3A and WNT-16B as indicated. **B.** Kinetic Δ BRET traces and average Δ BRET values of HEK293 cells transfected with FZD₅-Nluc, LRP6-Venus, and RNF43 (wt and the dominant-negative R286W and D300G mutants),

stimulated with 1000 ng/ml WNT-16B. **C.** Kinetic Δ BRET traces and average Δ BRET values of HEK293 cells transfected with FZD₅-Nluc, LRP6-Venus, and either pcDNA or wt RNF43. Cells were pre-incubated for 2 h with either vehicle or 100 ng/ml R-spondin 1 and subsequently stimulated with 1000 ng/ml WNT-16B. Data are shown as mean \pm SEM of three or four independent experiments (see on the bar graph on the right side), each performed in triplicate. Average Δ BRET values were analyzed with a one-way ANOVA followed by Sidak's multiple comparisons, where the mean of each group is compared with the mean of every other group; *p*-values are displayed in the figure for comparisons of interest.

9. In Figure 3, the Topflash assays were performed after 24 hours of Wnt stimulation, whereas the BRET assays were performed within 60 minutes. The authors should include staining for phosphorylated β -catenin and total β -catenin within the same timeframe.

We monitored BRET between FZD₅ and LRP6 in response to WNT treatment for 60 min after treatment, as we can achieve a stable BRET signal across this period and not much longer. BRET assays monitor changes in real time and have reached their maximum approximately after 10 min, demonstrating that association between FZD₅ and LRP6 is a relatively rapid process. We observed similar timings in single molecule tracking experiments.

The kinetics of LRP6 phosphorylation, the defining proximal downstream event of WNT/ β -catenin signaling, have previously been investigated in WNT-3A-stimulated HEK293 cells¹⁶. Faint P-LRP6 bands could be observed after 15 min – where the BRET signal had already reached a maximum. The strongest P-LRP6 signal was detected after 3h, and it decreased 6h after WNT stimulation. We have included analyses of total β -catenin accumulation and β -catenin phosphorylation – readouts surely downstream of LRP6 phosphorylation – after 2 h post WNT addition, the time point we used for all immunoblotting experiments and detected an increased amount of total β -catenin and a strong decrease of phosphorylated β -catenin upon WNT-3A, but not upon WNT-16B treatment (Supp. Fig. 6B-E). Resolving the kinetics of different hallmarks of the WNT/ β -catenin signal branch is beyond the scope of this paper, whose focus is on the direct comparison of WNT-16B and WNT-3A and their effect on WNT receptor clustering.

TOPFlash reporter gene assays on the other hand, reflect on expression of an exogenous reporter gene (Firefly luciferase) in response to β -catenin stabilization, or more precisely TCF/LEF activation. This is on the one hand a far slower, more downstream process, and on the other hand we measure the accumulated reporter gene expression over the whole duration of 24 h. This is largely uncoupled from the state of both receptor complexation and β -catenin phosphorylation/accumulation at the time

of measurement. A direct kinetic comparison would be misleading due to the fundamental differences in what these assays measure.

References

1. Schulte, G. et al. Purified Wnt-5a increases differentiation of midbrain dopaminergic cells and dishevelled phosphorylation. *J. Neurochem.* **92**, 1550–1553 (2005).
2. Petersen, J. et al. Agonist-induced dimer dissociation as a macromolecular step in G protein-coupled receptor signaling. *Nat. Commun.* **8**, (2017).
3. Kilander, M. B. C., Halleskog, C. & Schulte, G. Recombinant WNTs differentially activate β -catenin-dependent and -independent signalling in mouse microglia-like cells. *Acta Physiol.* **203**, 363–372 (2011).
4. Halleskog, C. & Schulte, G. Pertussis toxin-sensitive heterotrimeric Gai/o proteins mediate WNT/ β -catenin and WNT/ERK1/2 signaling in mouse primary microglia stimulated with purified WNT-3A. *Cell. Signal.* **25**, 822–828 (2013).
5. Schihada, H., Kowalski-Jahn, M., Turku, A. & Schulte, G. Deconvolution of WNT-induced Frizzled conformational dynamics with fluorescent biosensors. *Biosens. Bioelectron.* **177**, 112948 (2021).
6. Grätz, L., Voss, J. H. & Schulte, G. Class-Wide Analysis of Frizzled-Dishevelled Interactions Using BRET Biosensors Reveals Functional Differences among Receptor Paralogs. *ACS Sens.* (2024) **9**, 9
7. Zhao, S., Wan, X., Dai, Y., Gong, L. & Le, Q. WNT16B enhances the proliferation and self-renewal of limbal epithelial cells via CXCR4/MEK/ERK signaling. *Stem Cell Rep.* **17**, 864–878 (2022).
8. Movérare-Skrtic, S. et al. Osteoblast-derived WNT16 represses osteoclastogenesis and prevents cortical bone fragility fractures. *Nat. Med.* **20**, 1279–1288 (2014).
9. Kaykas, A. et al. Mutant Frizzled 4 associated with vitreoretinopathy traps wild-type Frizzled in the endoplasmic reticulum by oligomerization. *Nat. Cell Biol.* **6**, 52–58 (2004).
10. Nile, A. H. et al. A selective peptide inhibitor of Frizzled 7 receptors disrupts intestinal stem cells article. *Nat Chem. Biol.* **14**, 582–590 (2018).
11. Kowalski-Jahn, M. et al. Frizzled BRET Sensors Based on Bioorthogonal Labeling of Unnatural Amino Acids Reveal WNT-Induced Dynamics of the Cysteine-Rich Domain. *Sci. Adv* **7**
12. Strakova, K. et al. The tyrosine Y2502.39 in Frizzled 4 defines a conserved motif important for structural integrity of the receptor and recruitment of Disheveled. *Cell. Signal.* **38**, 85–96 (2017).

13. Kilander, M. B. C., Dahlström, J. & Schulte, G. Assessment of Frizzled 6 membrane mobility by FRAP supports G protein coupling and reveals WNT-Frizzled selectivity. *Cell. Signal.* **26**, 1943–1949 (2014).
14. Jiang, X., Charlat, O., Zamponi, R., Yang, Y. & Cong, F. Dishevelled promotes wnt receptor degradation through recruitment of ZNRF3/RNF43 E3 ubiquitin ligases. *Mol. Cell.* **58**, 522–533 (2015).
15. Bugter, J. M. et al. E3 ligases RNF43 and ZNRF3 display differential specificity for endocytosis of Frizzled receptors. *Life Sci Alliance* **7**, (2024).
16. Khan, Z., Vijayakumar, S., de la Torre, T. V., Rotolo, S. & Bafico, A. Analysis of Endogenous LRP6 Function Reveals a Novel Feedback Mechanism by Which Wnt Negatively Regulates Its Receptor. *Mol Cell Biol.* **27**, 7291–7301 (2007).

Reviewer #3 (Remarks to the Author):

Bold green text indicates where changes in the manuscript were made.

We thank the reviewer for an interesting and giving discussion. The field suffers from the need for generalized models and the inability to embrace the complexity of a very complex signaling system where ten FZDs, 19 WNTs and several coreceptors feed into the WNT signaling system. The dogma of WNT/ β -catenin signaling is tied to signalosome formation and is not adapted to potential exceptions. In our work we have discovered discrepancies in WNT behavior related to receptor engagement and signaling output. Surely, questions remain and require thorough experimental follow-up to clarify the molecular determinants that define ligand-receptor binding selectivity, receptor activation and signal initiation. We fully understand the reviewer's criticism and the relevant questions that were raised. We hope that the following argumentation can convince the reviewer that the data presented in our manuscript fully supports our conclusion even though they might not fully explain the mode of action of WNT-16B in physiology. Furthermore, we hope to convince this reviewer about the novelty of our data.

The main novelties in our work are:

1. WNT-16B leads to FZD-LRP5/6 association in BRET assays (WNT-5A does not do that in our hands) and elicits an electrophoretic mobility in DVL but is not able to initiate a WNT/ β -catenin signaling cascade. This correlates with its inability to establish higher order receptor clusters and provides a counterinstance to the prevailing model in the field stating that ligand-induced interaction of FZD and LRP6 necessarily is followed by WNT/ β -catenin signaling. As reflected in the title, we consider this finding to be highly relevant for the field and believe that this is the core finding of the manuscript.
2. We deliver direct evidence for the order of events in FZD-LRP6 clustering and LRP6 phosphorylation upon WNT-3A stimulation. This has, as pointed out by the reviewer, been touched upon by Bilic et al., who employed overexpression of a dominant-negative CK1 γ isoform, however their data leaves plenty of room for interpretation (see below).
3. The presence of DVL in the cell changes the orientation of FZD and LRP6 C-termini towards each other.

The reviewer appreciates the additional data the authors have provided to clarify the technical aspects of this study. However there remain concerns about whether the overarching conclusions are fully supported by the data presented in this study.

We are happy that the reviewer appreciates our data addressing the more technical concerns that were raised in the first round of review. It should be underlined that the reviewer's concerns "about whether the overarching conclusions are fully supported by the data presented in this study" are entirely new and were not mentioned in the first round of review.

The authors present the central finding that WNT16b induces the association of FZD5 and LRP6 but does not activate β -catenin-dependent signaling. They propose that WNT16b challenges the idea that β -catenin activation necessarily follows from ligand-induced FZD-LRP6 interactions, an assumption commonly supported by the use of WNT surrogates, which act as extracellular crosslinkers to potently trigger WNT/ β -catenin signaling.

Conceptually, however, this finding is not entirely novel. Previous reports have demonstrated that non-canonical WNTs (e.g., WNT5a) can engage LRP6 without activating β -catenin mediated signaling (Sato et al., 2009, EMBO J.; Bryja et al., 2009, Mol. Biol. Cell).

While WNT-5A binding to LRP6 has been shown by pull down and biochemical means in previous publications, functional relevance of this interaction for the initiation of signaling events is non existing or weak. This is in agreement with our experiments showing that WNT-5A is not able to bring FZDs and LRPs together. For WNT-16B on the other hand, the literature (as this reviewer indicated in comment 1 of the first revision) supports the concept that both β -catenin-dependent and -independent pathways can be activated (Moverare-Skrtic et al., 2014; Sun et al., 2012; Teh et al., 2007; Zhao et al., 2022). This distinguishes WNT-5A clearly from WNT-16B and poses in part the basis for our work.

We have emphasized the differences between WNT-5A and WNT-16B in the result section.

The authors provide extensive evidence showing that WNT16b does not induce LRP6 phosphorylation or β -catenin-driven transcription, consistent with earlier findings indicating that WNT16b engages non-canonical pathways (Teh et al., 2007, J. Cell Sci.).

The BRET assay showed interaction for WNT16b-LRP6 but not WNT5a-LRP6. It is not clear why the two non-canonical ligands behave differently.

Indeed, WNT-16B does not activate pathway-specific hallmarks of WNT/ β -catenin signaling in our hands (except for bringing FZDs and LRP5/6 together). However, as pointed out by this reviewer in the first revision (comments #1), WNT-16B was also reported to mediate WNT/ β -catenin signaling and thus our initial assumption was that WNT-16B activates both β -catenin-dependent and -independent signaling (Moverare-Skrtic et al., 2014).

It must be underlined that we cannot extract any direct information of WNT-5A binding to LRP6 alone because the employed readout is designed to monitor FZD-LRP6 proximity, not LRP6 binding per se. Thus, we can only state that WNT-5A does not lead to FZD-LRP5/6 association. We would like to emphasize the possibility that WNTs that do not activate WNT/ β -catenin signaling can act differently, one non-canonical WNT does not need to activate the same array of downstream signaling as another non-canonical WNT.

It is important to note that HEK293 cells have high endogenous LRP6/FZD and are typically used for β -catenin mediated TOPFlash activity. Therefore, detecting a WNT16b-LRP6 interaction is not completely unexpected and does not necessarily reflect a broader physiological context.

WNT/ β -catenin signaling is initiated and specified by WNT binding to FZDs and LRP5/6. WNT-3A is very good at this, WNT-5A is not. The molecular underpinnings for this dichotomy remain unclear even though some progress has been made by creation of chimeric WNTs (Tsumumi et al., 2023). As mentioned in the manuscript and by the reviewer in the comment #1 of the first review, WNT-16B elicits β -catenin-dependent and -independent signaling (Moverare-Skrtic et al., 2014; Teh et al., 2007). While the differences might be cell type dependent, WNT-16B clearly interacts with FZD and LRP5/6 to form a heterodimeric receptor complex ("association"), very similar to what WNT-3A can accomplish. We therefore agree with the reviewer, that LRP5/6 interaction of WNT-16B is not surprising at all. What is surprising, however, is the inability of

WNT-16B - in contrast to WNT-3A - to initiate a full blown WNT/ β -catenin cascade, while both act similar in the FZD-LRP5/6 BRET assay monitoring receptor association.

As a side note, we do not understand the reviewer's argument about the high endogenous expression levels of LRP6 and FZDs. How would that interfere with any of the experiments that we have performed, especially underlining that we employ labelled receptor constructs or cells devoid of FZDs or devoid of LRP5/6 in suitable experimental settings. Despite the presence of endogenous receptors, different WNT proteins (3A, 5A, 16B) exhibited distinct activity profiles. WNT-3A and WNT-5A behaved as expected, whereas WNT-16B fully activated upstream readouts. According to the signalosome model, it should therefore potently activate WNT/ β -catenin signaling, but it did not. This in fact is the core of paper.

In this context it is interesting to compare the WNTs by protein sequence, similar to what was done by Tsutsumi and colleagues (Tsutsumi et al., 2023). Here, the authors determined a section termed NC-linker as a molecular determinant of WNT-LRP6 engagement. This linker is one of the least conserved amino acid stretches among WNT paralogs but conserved between species (see below for a sequence alignment of human WNT paralogs). WNT-1 and WNT-3/3A have fully different NC-linkers, resembling binding to other propellers. WNT-5A and WNT-16B have shorter NC-linkers, which otherwise share almost no homology outside of two N-terminal cationic residues. Otherwise, the WNT-16B linker is largely basic, similar to the NC-linkers of LRP-engaging WNTs.

As the sequence difference in this decisive region is likely accompanied by another molecular mechanism of WNT-LRP6 binding, this might speculatively hint at a different complex architecture of the WNT-16B-LRP6-FZD complex when compared to a WNT-3A-bound complex. Nevertheless, it is contradicting the signalosome model that WNT-16B, while clearly engaging both FZD and LRP6 *and* leading to association of both (which WNT-5A does not do) in live cells, does not result in productive WNT/ β -catenin signaling. This is the key novelty of this manuscript.

WNT-16	-----MDRAALLGLARLCLAW-----AALLVLFY--GAQGNMMLGIASF-----	39
WNT-5A	MKKSIGILSPGVALMGAGSA--MSSKFFLVALAIFFSFAQVVIANSWWSLGMNPNVQMS	58
WNT-1	-----MGLWALLPGWVSATLLLA--LAALPAALAANS SGRWWGIVNVASSTNL	46
WNT-3A	-----MAPL--GY----F-LL-LCS--LKQALGSYPIWWSLAVGQPQYSSL	35
WNT-3	-----MEPHLLGL---LLGL-LLG--GTRVLAGYPIWWSLALGQQYTSL	38
	: * .:	
WNT-16	----GVPEKLGCANLP-LNSRQKELCKRKPYPYLLPSIREGARLGIQECGSQFRHRERWNCM	93
WNT-5A	EVYII GAQ--PLCSQLAGLSQGGKKLCHLYQDHMQYIGEGAKTGIKECQYQFRHRRWNC	116
WNT-1	LTDSKSLQL-VLEPSLQLLSRKQRRLIRQNPGLHVSVGGGLQSAVRECKWQFRNRWNC	105
WNT-3A	----GSQP-ILCASIPGLVFKQLRFRCRNYEIMPVAEGIKIGIQECQHQRFRRRWNC	89
WNT-3	----GSQP-LLCGSIPGLVFKQLRFRCRNYEIMPVAEGVKGIGIQECQHQRFRRRWNC	92
	. : * * . : : * : : * * * * . * * * *	
WNT-16	ITAAATTAPMGASPLFGYELSSGTKETAFIYAVMAAGLVHSVTRSCSAGNMTECSDTTL	153
WNT-5A	TVD-----N-TSVFGRVMQIGSRETAFTYAVSAAGVNVMSRACREGELSTCGCSRAA	168
WNT-1	TAP-----G-PHLFGKLVNRGCRETAIFAITSAGVTHSVARSCEGSIESCDCYRR	157
WNT-3A	TVH-----DSLAI FGPVLDKATRESAFVHAIASAGVAFVTRSCAEGTAAICGSSRH	142
WNT-3	TID-----DSLAI FGPVLDKATRESAFVHAIASAGVAFVTRSCAEGTSTICGCDSHH	145
	. : * * . : : * * * * . : : * * * * * * *	
WNT-16	QNGGSASEGWHWGGCSDDVQYGMWFSRKFLDFPIGNT---TGKENKVLAMNHNNEAGR	210
WNT-5A	R-PKDLPRDVLWGGCGDNIDYGYRFAKEFVDARERERIHAKGSYESARILMNLHNNEAGR	227
WNT-1	R-G-PGGPDWHWGGCSDNIDFGRLFGREFVDSGEKGR-----DLRFLMNLHNNEAGR	207
WNT-3A	Q-G-SPGKGWKGCCSEDI EFGMVSRFADARENRP-----DARSAMNHRHNNEAGR	192
WNT-3	K-G-PPGEGWKWGGCSEADDFGLVLSRFADARENRP-----DARSAMNKHNEAGR	195
	: * * * * . : : * * . : : * * * * * * * * * * * * * * * * * *	
WNT-16	QAVAKLMSVDCRCHGVSGCAVTKCWKTMSSFEKIGHLLKDKYENS IQISDKTKRK---	266
WNT-5A	RTVYNLADVACKCHGVSGCSLRTCWLQLADFRKVGDALKEKYDSAAMRINSRGGK---	283
WNT-1	TTVFSEMRQECKCHGMSGCTVRTCWMRLPTLRAVGDVLRDRFDGASRVLYGNRGSNRRAS	267
WNT-3A	QAIAASHMLKCKCHGLSGSCEVKTCCWWSQPDFRAIGDFLKDKYDSASEMVEKH---RES	249
WNT-3	TTILDHMLKCKCHGLSGSCEVKTCCWWSQPDFRAIGDFLKDKYDSASEMVEKH---RES	252
	: : . * : * * : * * * * : * * * * : : * * * * * : : * * * * * : : *	
WNT-16	----MRRREKDKQKIPHKD DLYVNVKSPNYCVEDKKLGIPTGQGRECNRTSEGADGCNL	322
WNT-5A	-----LVQVNSRFNSPTTQDLVYIDPSPDYCVRNSTGSLGTQGRLCNKTSEGMDGCEL	337
WNT-1	RAELRLLEPEDPAHKPPSPHDLVYFEKSPNFCTYSGRLGTAGTAGRACNSSSPALDGCCEL	327
WNT-3A	RGWVETLRPRYTYFRVPTERDLVYIEASPNFCEPNPETGSGFGRDRTCNVSSHGIDGCCL	309
WNT-3	RGWVETLRRAKYSLFKPPTERDLVYIENSFPNFCENPETGSGFGRDRTCNVSSHGIDGCCL	312
	* * * * : * * * * . * * * * * * * * * * * * * * * * * *	
WNT-16	LCCGRGYNTHVVRHVERCECKFIWCCYVRCRRCESMTDVHTCK	365
WNT-5A	MCCGRGYDQFKTVQTERCHCKFWCCYVKCKKCTEIVDQFVCK	380
WNT-1	LCCGRGHRTRTQRVTERCNCFTFWCCHVSCRNCTHTRVLHECL	370
WNT-3A	LCCGRGHNARAERRREKRCRVFWCCYVSCQECTRVYDVHTCK	352
WNT-3	LCCGRGHNTRTEKRKEKCHCIFHWCCYVSCQECIRIYDVHTCK	355
	: * * * * : * * * * * * * * * * * * * * * * * *	

Furthermore, the authors also indicated the overexpression of FZD/LRP lead to the high levels of surface expression, and lead to the saturation of ubiquitination by RNF43.

Indeed, we added additional experiments with RNF43 and RSPO1 to the revised version (revision #1) of our manuscript. We are happy that the reviewer appreciates the data.

The nature of LRP6 and WNT16b interaction reported here requires further clarifications. Can WNT16b signal in the absence of LRP6? Could WNT16b interaction with LRP6 stabilized by other cofactors, not present in HEK293 cells? In previous studies involving Wnt16b, other cell types have been used.

This manuscript focuses on the initiation of WNT/ β -catenin signaling and we put WNT-16B in the spotlight as it is a counterinstance for the widespread assumption that association of FZD and LRP6 is both sufficient and necessary for the initiation of WNT/ β -catenin signaling. While we still consider it to be necessary, we do not consider it to be sufficient. Experiments in absence of LRP6 would not allow us to further elaborate on WNT/ β -catenin signaling as LRP6 is crucial for specification of this pathway. Other non-canonical signaling pathways are often poorly defined and lack proper readouts. As a substitute, we have measured conformational rearrangement of FZD and the DVL2 DEP domain in a unimolecular biosensor recently published by us (Gratz et

al., 2024). HEK293 cells and HEK293 Δ LRP5/6 cells showed no difference here, indicating that FZD/DVL activation is induced by WNT-16B independent of LRP6 (data are only presented here in the rebuttal letter).

Fig. 1: Knockout of Δ LRP5/6 does not affect FZD/DVL activation by WNT-16B. Δ BRET traces of HEK293 cells and Δ LRP5/6 cells transfected with a FZD₅-DEP-Clamp conformational biosensor, stimulated with WNT-16B (1000 ng/ml). Data points are average \pm SEM of three independent experiments performed in triplicate.

Thus, WNTs can act on FZDs in the absence of coreceptors such as LRP5/6. We have just published the following data showing that the best-described prototypic WNT, WNT-3A, activates a FZD-DEP Clamp sensor, a proxy for FZD-DVL communication independent from LRP5/6, too. Additionally, we have published many other experimental paradigms where we show that removal of LRP5/6 input by KO or DKK treatment does not affect WNT-induced and FZD-mediated effects (upstream or outside WNT/ β -catenin signaling) (Bowin et al., 2022; Gratz et al., 2024; Halleskog and Schulte, 2013; Kowalski-Jahn et al., 2021; Schihada et al., 2021).

At this point we would like to turn the question back to the reviewer: What information would we obtain from the experiments of removing LRP6 from the system and assessing WNT-16B efficacy, when our original hypothesis was that a WNT-16B-induced FZD/LRP complex is formed but does not feed into efficient WNT/ β -catenin signaling, whereas the most well established “canonical” WNT, WNT-3A, does this?

[figure redacted]

Fig. 2, From Grätz et al 2024 ACS Sensors (Gratz et al., 2024)

We agree with the reviewer's remark regarding a potential cofactor not present in HEK293 cells. Indeed, we raised the same hypothesis in our discussion section and clearly indicated that research ahead will clarify those and that the current dataset does not provide answers in this direction (from our discussion: "Speculatively, cell type-specific expression of a co-factor absent in HEK293/CHO cells might be required to initiate WNT/ β -catenin signaling upon WNT-16B stimulation." **We further elaborated on this point in the discussion to highlight this explanation for our results, which might provide a basis for further intriguing research projects identifying further factors impacting signal specification.** We hope that the reviewer understands that the identification and validation of such a factor is outside of the scope of this manuscript.

A major limitation of the overarching conclusion of this study 'WNT-induced association of Frizzled and LRP6 is not sufficient for the initiation of WNT/ β -catenin signaling' is primarily based on one Wnt ligand, i.e., WNT16b. In many cases, e.g., WNT3a, ligand-induced FZD-LRP6 interaction does lead to clustering and initiate robust β -catenin signaling.

First of all, the main criticism here is that we only use one ligand to put our findings forward. We do not agree with this argumentation: A general model (such as that FZD/LRP association is necessary and sufficient for the activation of WNT/ β -catenin signaling) can indeed be falsified by a single exception. In our hands WNT-16B provides such a counterinstance.

Also, we fear that nomenclature and/or semantics are the origin of confusion. Terms such as receptor-receptor interaction, heterodimerization, crosslinking, association, clustering and higher order clustering are used here and in the literature. **We have attempted to clearly distinguish between receptor association (increase of interaction propensity across the whole sample as measured in BRET) and higher order clustering (as assessed by single-molecule tracking). We went through the manuscript again for the sake of consistency in this terminology since it apparently led to misunderstandings.**

Similarly, WNT surrogates can crosslink these receptors and drive canonical WNT activity, even when acting through FZD isoforms thought to be non-canonical. This study did not investigate whether Wnt surrogate initiates higher order clustering, a critical gap in the analysis.

We aimed to study the endogenous FZD ligands of the WNT family, and thus we disagree that the absence of analysis using WNT surrogates presents a critical gap in our work. We used the surrogates in the context of control experiments, and we did not directly study the surrogates' ability to form higher order clusters. This has in fact been shown by single molecule tracking in two examples ((Janda et al., 2017) and <https://www.biorxiv.org/content/10.1101/2024.06.18.599024v1>). It should be underlined that the first surrogates were composed of a CRD binder and DKK providing 1:1 stoichiometry and that those induced higher order clusters, work that clearly inspired us technically.

In short, the requested experiments were already published by others.

The observation that "initial ligand-induced FZD-LRP6 association must be followed by receptor

clustering into higher-order complexes and subsequent phosphorylation of LRP6 for efficient activation” has been long recognized in the field (e.g., Bilic, Science, 2007).

While Bilic et al clearly presents a milestone paper, their work does not address WNT-induced FZD-LRP6 interaction experimentally. Instead, Bilic et al. report on the presence of non-phosphorylated, WNT-induced LRP6 aggregates in cells expressing a dominant-negative mutant of CK1 γ . We think that the data provided by Bilic et al. leaves a substantial room for interpretation, even though a strong statement is put forward in the publication.

Our single-molecule microscopy data employing the phosphorylation-deficient LRP6-5A mutant add direct evidence to the order of events: If phosphorylation was a prerequisite to higher-order LRP6 clustering, there would be no clusters, however, we observed higher-order clusters similar as we did when using a wild-type LRP6 construct. We would like to point out that our mutational paradigm allows for a more conclusive interpretation than co-transfection of dominant-negative CK1 γ :

- (i) CK1 γ (co)transfection efficiency was not controlled for in Bilic et al.
- (ii) The employed antibody in Bilic et al. only assesses phosphorylation of a specific residue (T1479), while the LRP6 C-tail has multiple phosphorylation motifs and residues whose phosphorylation might go unnoticed.
- (iii) Adding to that, other CK1 isoforms, GSK3 β , and unrelated kinases can contribute to LRP6 phosphorylation (Cervenka et al., 2011; Wolf et al., 2008) and other phosphorylatable residues not captured by the employed antibody might be substrates to kinases beyond CK1 γ .
- (iv) Bilic et al. observed residual T1479-phospho-LRP6 aggregates in the exemplary snapshot shown in Fig. 4C, emphasizing that the dominant-negative CK1 γ mutant did not abolish LRP6 phosphorylation. In our eyes, the LRP6-5A mutant provides a more compelling experimental paradigm.

In total, the data from Bilic et al. point in the direction of our findings, but do not provide waterproof evidence for the strongly forwarded conclusion. **We have expanded on this matter in the discussion section.** Since even recent reviews by long-standing experts in the field suggested an inverse order of events (Maurice and Angers, 2025): “*When receptor complex activation by Wnt binding promotes proximity between FZD and LRP5/6 receptors at the cell surface, the plasma membrane recruitment of a pool of the Axin–GSK3 complex leads to LRP5/6 phosphorylation within cytoplasmic PPPSPXS sequences priming for further phosphorylation by CK1 γ that provides priming for further phosphorylation by CK1 γ . This triggers the formation of LRP5/6 signalosomes, which exist as discrete puncta at the plasma membrane*” we think that our direct experimental evidence in this manuscript emphasizing that LRP6 phosphorylation in the conserved PPPS/TP motifs is not required for higher-order clustering will be appreciated by the field as it was by reviewer #2, and will help to guide our understanding of WNT/ β -catenin signaling initiation.

The author observed clustering with single particle tracking for WNT3a but not WNT16b, which also fail to activate b-catenin signaling. The study did not directly show that monomeric WNT3A-Lrp6-Fzd interaction is insufficient for b-catenin signaling. Rather the conclusions are largely based on the observation that Wnt16b that interact with LRP6 but did not form higher order clusters or activate WNT signaling.

We agree with the reviewer that an experimental paradigm that would halt receptor interactions at the level of 1:1 association preventing higher order clustering would be an attractive

approach. We cannot come up with an experimentally feasible approach to capturing a monomeric WNT-3A-LRP6-FZD complex since WNT-3A rapidly induces higher-order FZD-LRP6 clustering in our hands.

Nevertheless, we present the effect of WNT-16B to exactly support this point since WNT-16B does associate the receptor but does not cluster them in higher order cluster, as the reviewer pointed out. It should be again underlined that WNT-16B physically interacts with both FZD and LRP6 similar to WNT-3A but unlike WNT-5A (which in our BRET experiments does not bring LRP6 and FZDs together despite overwhelming evidence that it interacts with the FZDs).

Other comments:

The RNA-seq data for both Wnt3a and Wnt16b are somewhat inconclusive. The authors claimed that HEK293 cells may not be optimal for studying WNT signaling but this is puzzling. RNA-seq data and other experiments such as single particle tracking were not performed in the presence of R-spondin, even though the TOP-flash assay showed marked increase in activity in the presence of WNT3a/R-spondin. Likewise, HEK293 may not be optimal for studying β -catenin independent signaling.

As discussed in the first revision, we are aware of the limitations of the RNA-seq data. While we still think that the dataset is suitable to illustrate that WNT-3A and WNT-16B lead to both shared and specific responses, we concluded that HEK293 cells may not be an ideal model to assess the transcriptional changes caused by WNT/ β -catenin signaling as (i) established target genes were not upregulated by WNT-3A, and (ii) there is probably a high basal WNT/ β -catenin signaling activity (see previous revision). We have stated these limitations clearly in the text and forwarded our arguments carefully.

While it is well-established that RSPO1 further boosts WNT/ β -catenin signaling (exemplified by our TOPFlash results, as the reviewer pointed out), we opted to keep our experimental paradigms as simple as possible. Since treatment with WNT-3A alone caused a significant increase in TOPFlash, and treatment with WNT-16B + RSPO1 did not, we considered treatment with WNT-3A alone sufficient for the RNA-seq samples. In single-molecule tracking, we did not consider using RSPO1 as its effect is mainly mediated by stopping RNF43/ZNRF1-dependent FZD internalization and degradation. In the short time scale of the respective experiments, we still see marked clustering of FZD₅ and LRP6 for WNT-3A-treated cells, but not for LRP6-treated cells, therefore a signal window also exists in absence of RSPO1. As RSPO1 treatment did not elevate the TOPFlash signal for WNT-16B treated cells, we have no reason to assume that it changes the molecular architecture of FZD-LRP-WNT-16B complexes, particularly when it does not directly interact with any of these proteins.

Regarding the usage of HEK293 cells as a model system: Easily transfectable cell systems present a reductionistic system to study molecular events in cellular signaling. In its simplicity, we think that conclusions can be drawn from direct molecular read outs, co-transfection of suitable receptors, and the addition of recombinant WNTs that differ in their activity profile. In this case, our work refines the chain of events in WNT-induced β -catenin signaling and elaborates on the ligand-selective induction of higher order complexes relevant for efficient signal initiation. These conclusions can be drawn from this simplistic cell system and would not have been possibly dissected in more complex cellular models. Future work must bring those

aspects closer together.

References

- Bowin C-F, Kozielwicz P, Grätz L, Kowalski-Jahn M, Schihada H and Schulte G (2022) WNT stimulation induced conformational dynamics in the Frizzled-Dishevelled interaction. *bioRxiv*:2022.2007.2019.500578.
- Cervenka I, Wolf J, Masek J, Krejci P, Wilcox WR, Kozubik A, Schulte G, Gutkind JS and Bryja V (2011) Mitogen-activated protein kinases promote WNT/beta-catenin signaling via phosphorylation of LRP6. *Mol Cell Biol* **31**:179-189.
- Gratz L, Voss JH and Schulte G (2024) Class-Wide Analysis of Frizzled-Dishevelled Interactions Using BRET Biosensors Reveals Functional Differences among Receptor Paralogs. *ACS Sens* **9**:4626-4636.
- Halleskog C and Schulte G (2013) Pertussis toxin-sensitive heterotrimeric G(alpha_i/o) proteins mediate WNT/beta-catenin and WNT/ERK1/2 signaling in mouse primary microglia stimulated with purified WNT-3A. *Cellular signalling* **25**:822-828.
- Janda CY, Dang LT, You C, Chang J, de Lau W, Zhong ZA, Yan KS, Marecic O, Siepe D, Li X, Moody JD, Williams BO, Clevers H, Piehler J, Baker D, Kuo CJ and Garcia KC (2017) Surrogate Wnt agonists that phenocopy canonical Wnt and beta-catenin signalling. *Nature* **545**:234-237.
- Kowalski-Jahn M, Schihada H, Turku A, Huber T, Sakmar TP and Schulte G (2021) Frizzled BRET sensors based on bioorthogonal labeling of unnatural amino acids reveal WNT-induced dynamics of the cysteine-rich domain. *Sci Adv* **7**:eabj7917.
- Maurice MM and Angers S (2025) Mechanistic insights into Wnt-beta-catenin pathway activation and signal transduction. *Nature reviews Molecular cell biology*.
- Moverare-Skrtic S, Henning P, Liu X, Nagano K, Saito H, Borjesson AE, Sjogren K, Windahl SH, Farman H, Kindlund B, Engdahl C, Koskela A, Zhang FP, Eriksson EE, Zaman F, Hammarstedt A, Isaksson H, Bally M, Kassem A, Lindholm C, Sandberg O, Aspenberg P, Savendahl L, Feng JQ, Tuckermann J, Tuukkanen J, Poutanen M, Baron R, Lerner UH, Gori F and Ohlsson C (2014) Osteoblast-derived WNT16 represses osteoclastogenesis and prevents cortical bone fragility fractures. *Nature medicine* **20**:1279-1288.
- Schihada H, Kowalski-Jahn M, Turku A and Schulte G (2021) Deconvolution of WNT-induced Frizzled conformational dynamics with fluorescent biosensors. *Biosens Bioelectron* **177**:112948.
- Sun Y, Campisi J, Higano C, Beer TM, Porter P, Coleman I, True L and Nelson PS (2012) Treatment-induced damage to the tumor microenvironment promotes prostate cancer therapy resistance through WNT16B. *Nature medicine* **18**:1359-1368.
- Teh MT, Blaydon D, Ghali LR, Briggs V, Edmunds S, Pantazi E, Barnes MR, Leigh IM, Kelsell DP and Philpott MP (2007) Role for WNT16B in human epidermal keratinocyte proliferation and differentiation. *J Cell Sci* **120**:330-339.
- Tsutsumi N, Hwang S, Waghray D, Hansen S, Jude KM, Wang N, Miao Y, Glassman CR, Caveney NA, Janda CY, Hannoush RN and Garcia KC (2023) Structure of the Wnt-Frizzled-LRP6 initiation complex reveals the basis for coreceptor discrimination. *Proc Natl Acad Sci U S A* **120**:e2218238120.
- Wolf J, Palmby TR, Gavard J, Williams BO and Gutkind JS (2008) Multiple PPPS/TP motifs act in a combinatorial fashion to transduce Wnt signaling through LRP6. *FEBS letters* **582**:255-261.
- Zhao S, Wan X, Dai Y, Gong L and Le Q (2022) WNT16B enhances the proliferation and self-renewal of limb epithelial cells via CXCR4/MEK/ERK signaling. *Stem Cell Reports* **17**:864-878.